# Contribution of lakes in sustaining the Sahara greening during the mid-Holocene

Yuheng Li[1], Kanon Kino[1], Alexandre Cauquoin[2] and Taikan Oki[1]

[1]Department of Civil Engineering, Graduate School of Engineering, University of Tokyo, Tokyo, Japan
[2]Institute of Industrial Science, University of Tokyo, Kashiwa, Japan

*Correspondence to*: Yuheng Li (yuheng@rainbow.iis.u-tokyo.ac.jp)

**Abstract.** The climate impact contribution of lakes to sustain the Green Sahara in the mid-Holocene (MH, 6000 years ago) is still under debate. To assess the lake-induced climate response over North Africa, we investigated the roles of Western Sahara lakes and Megalake Chad using reconstructions of MH Sahara lake maps as surface boundary conditions for the

isotope-enabled atmospheric model MIROC5-iso. Our results show that the Western Sahara lakes pushed the West African monsoon northward and extended it eastward by expanding Megalake Chad. This lake-climate impact was caused by the cyclonic circulation response related to the weakened African Easterly Jet and enhanced Tropical Easterly Jet. According to the Budyko aridity index, the northwestern Sahara climate region shifted from hyper-arid to arid or semi-arid with the lake expansion. Moreover, precipitation scarcity could have been reduced by up to 13% to sustain the semi-humid conditions.

Such lake-climate impacts could alleviate Sahara aridity, relying on lake positions in the monsoon regions. Our findings are promising for understanding the contribution of lakes to sustaining the Green Sahara.

# 1 Introduction

Paleoclimatic evidence suggests that the Sahara, the largest hot desert in the world, was much wetter and greener during the mid-Holocene (MH, 6000 years ago) than in the present (Gasse, 2000; Adkins et al., 2006; Claussen et al., 2017). This period, called the Green Sahara (GS) or African Humid Period (AHP), was primarily caused by the Earth's orbital cycle revolution on obliquity, eccentricity, and precession, leading to high seasonality insolation in the Northern Hemisphere (Otto-Bliesner et al., 2017), with approximately 7% higher summer insolation over North Africa (NAf) during the MH compared with today (Berger, 1988). Under such orbital forcing changes, the West African Monsoon (WAM) strengthened and extended northward, leading to distinct rainfall regimes and increased vegetation along with narrow desert zones in the Sahara (Kutzbach et al., 2020). Although the GS climate is highly correlated with orbital forcing, state-of-the-art general circulation models (GCM) cannot account for the widespread precipitation during the GS period (Braconnot et al., 2007; Perez-Sanz et al., 2014; Harrison et al., 2015; Brierley et al., 2020). Hence, researchers have investigated oceanic and terrestrial roles in sustaining the GS. Remote oceanic impact contributes to enhanced summer monsoon with increasing sea surface temperature (Braconnot et al., 1999; Kutzbach and Liu, 1997; Zhao et al., 2005; Joly and Voldoire, 2009) and winter rainfall (Cheddadi et al., 2021) over NAf. Moreover, the inland terrestrial system is affected by vegetation growth (Thompson et al., 2022), especially interactions with soil (Kutzbach et al., 1996; Chen et al., 2020), dust reductions (Messori et al., 2018), and dust-cloud interactions (Hopcroft and Valdes, 2019; Braconnot et al., 2021). Despite terrestrial and ocean improvements in the model modules and an understanding of their roles in the GS climate, the MH climate simulations from the Paleoclimate Modeling Intercomparison Project 4 (PMIP4) still underestimate the northward WAM extension (Brierley et al., 2020).

Despite implementing all terrestrial impact in model MH simulations, biases still exist in the contribution of open-water surfaces (lakes and wetlands) over Naf, which are often set as the same as that in pre-industrial (PI) control simulations. Hoelzmann et al. (1998) reconstructed the Megalake Chad distribution in the Sahara during the Holocene (hereinafter small-lake map; LK_98 in Tables 1 and S1). By adopting this small-lake map to the Community Climate Model version 3, Broström et al. (1998) and Carrington et al. (2001) found that Megalake Chad produced more localized hydrological changes and did not contribute to the northward WAM movement. Contrastingly, using an improved atmospheric GCM, Krinner et al. (2012) further suggested that the open-water surface effect was underestimated in previous studies that reported the northward WAM shift, with a consequence of a doubling of the regional precipitation rates. However, the disadvantage of LK_98 is that it does not include any other MH Megalakes beyond Megalake Chad (Holmes and Hoelzmann, 2017). Hence, Chandan and Peltier (2020) further added dedicated MH Megalakes based on the small-lake map and investigated the lake effect using a fully coupled atmosphere-ocean GCM. They reported that the increase in precipitation from the lakes was weak, and the lake location did not considerably influence precipitation.

The contribution of Megalake Chad to the humidification of the Sahara is still under discussion. Furthermore, lakes in the Western Sahara also potentially contribute to the WAM. Tegen et al. (2002) further indicated the presence of larger lakes and wetlands over the Western Sahara based on dust emission simulations (hereinafter potential maximum-lake map; LK_02 in Tables 1 and S1). Based on LK_98 and LK_02, Specht et al. (2022) investigated the impacts of the latitudinal position of lakes and wetlands on changes in precipitation and initially highlighted the influence of western lakes on the northward WAM shift. These studies suggested that western lakes and Megalake Chad may play different roles in humidifying the Sahara; thus, this aspect requires further investigation.

The abovementioned studies on lake-climate impact also explored the underlying physical mechanisms by which lake thermal and dynamic forcing affects the atmospheric circulation of the African monsoon system. For example, compared with the enhanced localized water cycling forced by lake evaporation (Broström et al., 1998; Carrington et al., 2001), Krinner et al. (2012) considered that the cooling effect that stabilizes convection is only locally applicable to deep lakes but increases the precipitation rates in summer and delays cooling in autumn, thereby extending the monsoon. Recent studies have explored the mechanisms of how various components of the NAf monsoon system, including the Sahara Heat Low (SHL) and Sahara Highs in Western Sahara, African Easterly Jet (AEJ) in the middle atmosphere (600 hPa), and Tropical Easterly Jet (TEJ) in the upper atmosphere (200 hPa) influence the near-surface westerly flow northward and rainfall (Biasutti and Sobel, 2009; Claussen et al., 2017; Kuete et al., 2022). However, discrepancies exist regarding the effects of these components. Chandan and Peltier (2020) suggested that such a cooling effect could weaken the SHL and local convection, reducing the precipitation. Conversely, Specht et al. (2022) found that a weakened AEJ enhanced inland moisture transportation, leading to a northward and prolonged rain belt. Consequently, the mechanisms of lake-climate interaction in the NAf monsoon system remain unclear.

To address these issues, the present study assessed the contribution of Western Sahara lakes and Megalake Chad in humidifying the Sahara region during the MH using the isotope-enabled atmospheric GCM Model for Interdisciplinary Research on Climate version 5 (MIROC5)-iso (Okazaki and Yoshimura, 2019). To consider the large uncertainty in MH lake reconstructions (Quade et al., 2018), sensitivity experiments were conducted with the original two sets of lake reconstructions (Hoelzmann et al., 1998; Tegen et al., 2002) and recently updated high-resolution lake and wetland reconstructions maps (Chen et al., 2021) over the NAf during the MH. We discuss the influence of Western Sahara lakes and Megalake Chad on the WAM movement and the potential lake-climate mechanisms involved to sustain the GS.

## 2 Materials and Methods

### 2.1 Experiments and settings

We used the isotope-enabled version of the MIROC5 (Watanabe et al., 2010), hereafter called the MIROC5-iso (Okazaki and Yoshimura, 2019). The MIROC5-iso adopts a three-dimensional primitive equation in the hybrid σ–p coordinate system. Its resolution was set to a horizontal spectral truncation of T42 (approximately 280 km) and 40 vertical layers with

coordinates. The parameterization schemes have been comprehensively described by Watanabe et al. (2010), Okazaki and Yoshimura (2019), and Kino et al. (2021). The Minimal Advanced Treatments of Surface Interaction and Runoff (MATSIRO) model (Takata et al. 2003) is the MIROC land component, which simulates important water and energy circulation. The lake module simulates the thermal and hydrological processes of lakes and their interaction with the atmosphere. It sets a minimum lake depth threshold of 10 m, which means the lake permanently existed. Such isotope-enabled climate models have proven to be valuable tools for tracing water vapor transportation and identifying sources of precipitation changes (Tharammal et al., 2021; Liu et al., 2022).

To assess the hydroclimatic influence of the presence of lakes in NAf (0°–35° N; 20° W–40° E), we performed two control simulations for the PI (year 1850, $PI_{ref}$) and MH ($MH_{ref}$) periods and six MH sensitivity experiments (see Table 1). For every experiment, orbital forcing and greenhouse gas concentrations were set according to the PMIP4 protocol (Otto-Bliesner et al., 2017). Land surface boundary conditions (land-sea mask, ice sheets, soils, vegetation, and lakes) were set according to the Coupled Model Intercomparison Project Phase 5 (CMIP5) protocol for MIROC5-iso (Watanabe et al., 2010). Notably, the lake fraction is treated as the prescribed boundary conditions in the model based on the corresponding datasets because the model cannot dynamically simulate the lake. Specifically, the Earth Topography five-minute grid (ETOPO5, https://www.ngdc.noaa.gov/mgg/global/etopo5.HTML) was used as global lake map boundary conditions for the control simulations. In $MH_{ref}$ and $PI_{ref}$ experiments, the presence of lakes in NAf is minimal when using the global lake fraction map from the ETOPO5 in MIROC5-iso standard simulations (Fig. S1). In contrast, the other experiments show highly varied and much higher lake fractions. The distribution of vegetation types for all experiments can be observed in Fig. S2. Evidently, NAf is predominantly characterized by bare ground coverage. Each experiment was run for 60 y, and only the last 30 y were used to obtain the soil moisture (SM) of the study at an equilibrium state, which indicates a balanced land surface water budget, for our analyses. We used sea surface temperature, sea ice concentration, and water isotope content of the sea surface provided by MPI-ESM-wiso (Cauquoin et al., 2019) as boundary conditions for our PI and MH simulations. In our six sensitivity MH experiments, only the lake map was changed in NAf, while other boundary conditions were kept the same as in $MH_{ref}$ (Table 1).

Table 1. Experimental Settings

| Experiment | GHG + Orb | Sea Surface | Lakes in North Africa |
|---|---|---|---|
| $PI_{ref}$ | $PI^{*1}$ | $PI^{*2}$ | $ETOPO5^{*3}$ |
| $MH_{ref}$ | | | |
| $MH_C$ | | | $LK\_98^{*4}$ |
| $MH_{WC}$ | $MH^{*1}$ | $MH^{*2}$ | $LK\_02^{*4}$ |
| $MH_{WCE1}$ | | | $LK1^{*4}$ |
| $MH_{WCE2}$ | | | $LK2^{*4}$ |

| | |
|---|---|
| MH$_{WCE3}$ | LK3[*4] |
| MH$_{WCE4}$ | LK4[*4] |

*1 Following PMIP4 Protocol

*2 Cauquoin et al. (2019)

*3 National Geophysical Data Center, 1993. 5-minute Gridded Global Relief Data (ETOPO5) National Geophysical Data Center, NOAA. doi:10.7289/V5D798BF.

*4 Details of the lake reconstruction can be seen in Table S1.

The reconstructed lake maps in NAf that were used for our sensitivity experiments are summarized in Table S1 and shown in Fig. 1. MH$_C$ uses LK_98 (Hoelzmann et al., 1998), with only Megalake Chad, over 15°–20° E and 10°–20° N (Fig. 1a). The MH$_{WC}$ experiment uses LK_02 (Tegen et al., 2002), which has more western and northern lake areas over 0°–10° W and 10°–20° N in addition to Megalake Chad (Fig. 1b). The MH$_{WCE1}$–MH$_{WCE4}$ experiments use LK1–LK4 from Chen et al.

(2021), which show an increasing lake fraction in Megalake Chad and eastern lakes in South Sudan around 0–20° N, with a gradually increasing scattered west lake area (Fig. 1c–f) compared with LK_98 and LK_02. LK4 has the largest lake proportion in the western, eastern, and Megalake Chad regions, and differs from LK2 primarily in its representation of Megalake Chad (Fig. 1d and 1f). The average main lake fraction over the NAf region according to these different reconstructions varies from 1–13% compared with the total land areas of NAf (Fig. 1g). The water bodies delineated in

LK_98 and LK_02 only pertain to lakes, while LK1–LK4 include both wetlands and lakes. Generally, lakes and wetlands are persistently saturated or near-saturated areas that are regularly subjected to inundation or shallow water tables in the absence of human disturbances (Tootchi et al., 2019). In this study, wetlands were also treated as lakes in our climate model.

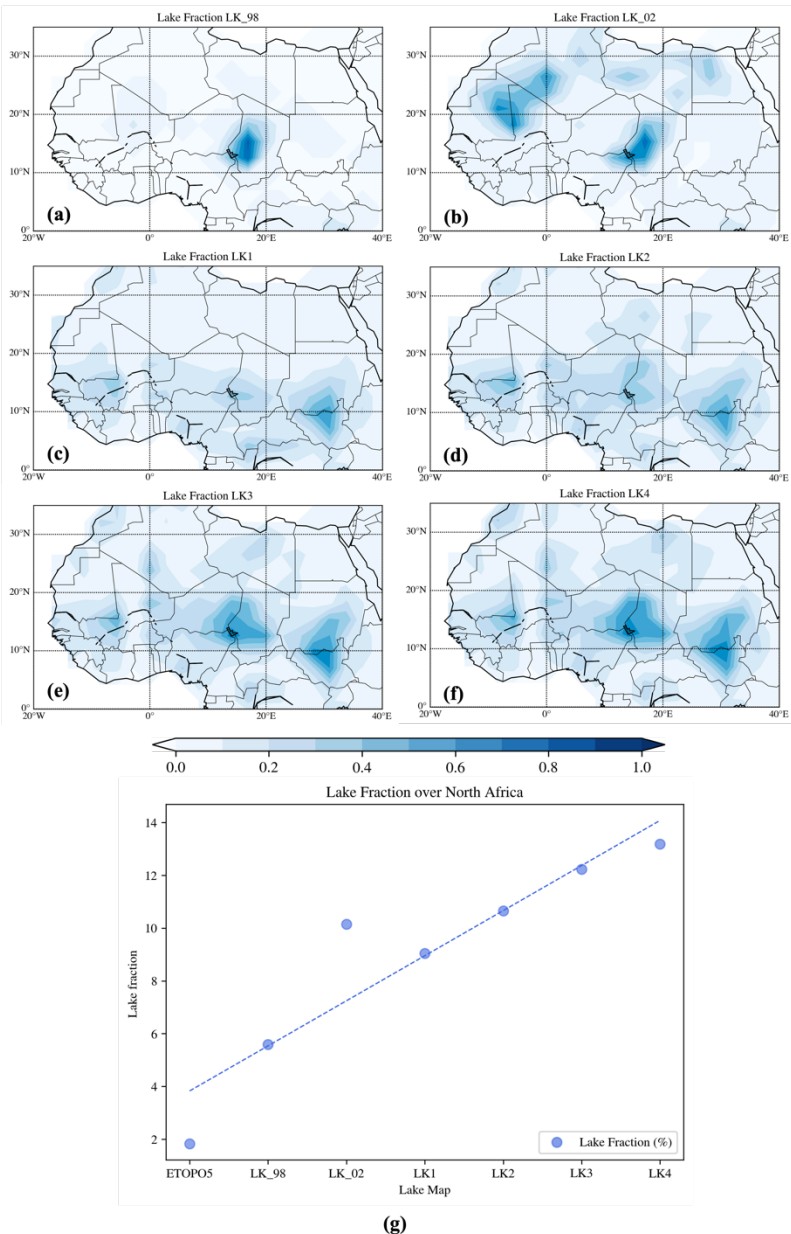

**Figure 1: Mid-Holocene (MH) lake maps in Northern Africa used in this study: (a) small lake map derived by Hoelzmann et al. (1998) used for the MH$_C$ experiments, (b) maximum lake map derived by Tegen et al. (2002) used for the MH$_{WC}$ experiments, (c)– (f) potential lake maps derived by Chen et al. (2021) corresponding to four different types of precipitation, used for the MH$_{WCE1}$, MH$_{WCE2}$, MH$_{WCE3}$ and MH$_{WCE4}$ experiments, respectively. The lake map differences mainly come from the Western Sahara lakes, Megalake Chad, and eastern lakes in South Sudan between 0°–20°N, and (g) fraction (circle size) of all the prescribed lake experiments compared with the total land areas of North Africa (0°–35° N; 20° W–40° E).**

We investigated the contribution of the Western Sahara lakes by comparing the MH$_C$ and MH$_{WC}$ experiments. The influence of Megalake Chad on NAf climate was assessed using the MH$_{WCE2}$ and MH$_{WCE4}$ results. To evaluate our model results, we compared the isotope outputs from MIROC5-iso with available observations from natural archives (e.g., $\delta^{18}$O in ice cores and speleothems) as in Cauquoin et al. (2019).

## 2.2 Climate model validation method

To evaluate our MH simulation, we used measured isotope datasets from ice cores and continental speleothems. We used five Greenland and ten Antarctic ice cores selected from the comprehensive compilations of Sundqvist et al. (2014) and WAIS Divide Project Members (Fudge et al., 2013), respectively. These are presented in Table 1 of Cauquoin et al. (2019). We also added to this dataset the MH−PI $\delta^{18}$O anomalies measured from four (sub)tropical ice cores (Huascaran, Sajama, Illimani, and Guliaa), as reported by Risi et al. (2010). Furthermore, we extracted 57 entities from the SISALv2 (Speleothem Isotope Synthesis and Analysis version 2) dataset (Comas-Bru et al., 2020), in which averaged $\delta^{18}$O values of calcite or aragonite are available for both the MH and PI period. As recommended by Comas-Bru et al. (2019), we defined the PI and MH as the means of the 1850–1990 CE and 6 ± 0.5 ka periods, respectively. The measured $\delta^{18}$O of calcite or aragonite were converted into $\delta^{18}$O of drip water using equations 1 or 2 of Comas-Bru et al. (2019), respectively, after converting V-PDB to the V-SMOW scale (equation 3 of Comas-Bru et al. (2019)). The annual mean surface air temperature from MIROC5-iso was used for the conversion.

## 2.3 Analysis methods

### 2.3.1 Hydroclimate analysis

We analyzed hydroclimate changes based on the ratio with the MH$_{ref}$ results.

$$Ratio_{exp} = \frac{Exp - MH_{ref}}{MH_{ref}} \times 100\% ,\tag{1}$$

The water vapor flux was also calculated to explain the precipitation changes. The zonal component of the vertically integrated flux (Fu) is:

$$F_u = \int_{300hpa}^{ps} \frac{uq}{g} dP ,\tag{2}$$

where $u$ is the zonal wind, $q$ is the specific humidity, $p$ is the pressure at a given vertical level, $g$ is the gravitational acceleration (9.8 m/s), and $ps$ is the surface pressure. The meridional component of the vertically integrated flux $Fv$ is expressed as:

$$F_v = \int_{300hpa}^{ps} \frac{vq}{g} dP\tag{3}$$

By combining $Fu$ and $Fv$, the integrated vapor transport can be expressed as:

$$IVT = \sqrt{F_u^2 + F_v^2}\tag{4}$$

## 2.3.2 Budyko aridity index

To assess climate zone transformation with the balance between available energy (net surface radiation) and water (precipitation) at the surface, the Budyko aridity index (Budyko and Miller, 1974) was calculated as a joint analysis using hydro-climatological variables as follows:

$$I = \frac{R_n}{lP} \tag{5}$$

where $R_n$ is the net surface radiation, $l$ is the latent heat coefficient ($2.5 \times 10^6$ J/kg), and $P$ is the precipitation at the surface. The change in the aridity index indicates regional shifts in hydroclimatic conditions.

The annual mean of net radiation and precipitation were used in the analysis. A higher Budyko aridity index indicates a drier region due to the high available energy relative to the amount of water, whereas a lower index indicates a more humid region due to the low available energy relative to the amount of water. In our study region, six climate regions were classified using the Budyko aridity index: tropical humid ($I \leq 0.7$), humid ($0.7 < I \leq 1.2$), semi-humid ($1.2 < I \leq 2.0$), semi-arid ($2.0 < I \leq 4.0$), arid ($4.0 < I \leq 6.0$), and hyper-arid ($6.0 < I$). The equation suggests that changes in the dryness index within a region are more indicative of shifts in the hydroclimatic regime over the long term rather than intra-annual variability, such as individual drought events.

## 3 Results

## 3.1 Model reproducibility

To evaluate the $MH_{ref}$ results at the global scale, we compared $MH_{ref}$ - $PI_{ref}$ with isotopic observations (Fig. S3). We found a reasonable model–data agreement, with root mean square error and R-squared values of $0.81 \times 10^{-3}$ and 0.33, respectively. In addition, MIROC5-iso simulated a decrease in the isotopic composition of precipitation over NAf due to the enhanced monsoon during MH, which agrees with previous model studies (Schmidt et al., 2007; Risi et al., 2010; Cauquoin et al., 2019). Our simulation bias mainly originated from the ice cores in Antarctica and speleothems in North America (Fig. S3a). The isotopic performances in the $PI_{ref}$ simulation were verified in Okazaki and Yoshimura (2019) and Kino et al. (2021). Other studies also confirmed the general reproducibility of global MH characteristics using the MIROC-series (O'Ishi and Abe-Ouchi, 2011; Ohgaito et al., 2021).

To further examine the model performance in NAf, we compare our precipitation result with the study conducted by Larrasoaña et al. (2013: Fig 4a). As shown in Fig. S4a, our results suggest that the performance of MIROC5-iso in reproducing the northward shift of the zone with precipitation <1000 mm/y could still be improved, although it shows good agreement with the reconstructed map in the zone with precipitation >1000 mm/y. Moreover, we also compared our result with precipitation and summer season temperature anomalies between 6–0 ka, as provided by Bartlein et al. (2010) (Fig. S4b–e). This comparison also revealed precipitation underestimation in northern NAf and lower temperatures in central NAf.

These comparisons collectively suggest a simulation bias of the MIROC5-iso model in NAf, particularly concerning the northward movement of the monsoon system.

We also examined the model representation of WAM characteristics (Claussen et al., 2017). Based on the annual cycle of WAM (Thorncroft et al., 2011), we defined summer as June-July-August-September (JJAS) and winter as January-February-March (JFM). We focused on summer because of the large amount of precipitation caused by WAM. In both MH$_{ref}$ and PI$_{ref}$,

the Sahara Highs in the middle atmosphere were positioned at 20°–30° N and centered at 0° E (contours in Fig. 2a and 2d). In the middle atmosphere, AEJ was found at 10°–15° N, corresponding to the precipitation belts (vectors and shaded areas in Fig. 2a and 2d), and the concurrent TEJ at 0°–10° N in the upper atmosphere (vectors in Fig. 2b and 2e). In the lower atmosphere (850 hPa), the SHL, centered in the hottest Sahara region at 10°–20° N (contours in Fig. 2c and 2f), was associated with the monsoon westerly winds from the equatorial Atlantic Ocean to the continent (vectors in Fig. 2c and 2f).

Although the monsoon westerly flow was at ~10° N in PI$_{ref}$, it moved to ~15° N in MH$_{ref}$. Because the model bias and uncertainty in reproducing the AEJ still require improvement in reanalysis datasets (Kuete et al., 2022), our climate model efficiently captured the WAM patterns for investigating the its sensitivity to lake expansions in the Sahara.

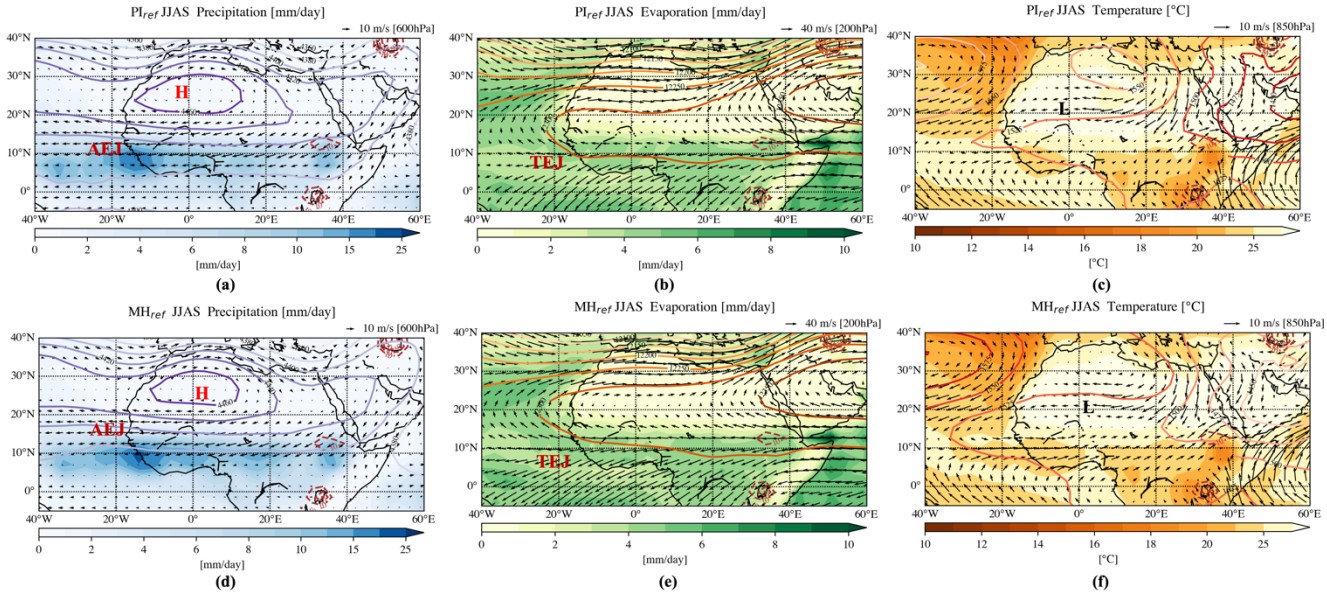

**Figure 2: Simulated climatological precipitation and temperature responses for the Pre-industrial (PI) and MH reference experiments during the summer season (June-July-August-September, JJAS). For the PI experiment, (a) is the precipitation with 600 hPa wind (arrow) geopotential height (counters), (b) is the evaporation with 200 hPa wind and geopotential height, and (c) is the surface temperature with 850 hPa wind and geopotential height. Subplots (d), (e), and (f) are the same as (a), (b) and (c), respectively, but for the MH experiment. For (a–f), lake fraction (%) contours of the respective lake sensitivity experiment are**
**shown with red dashed lines and the respective reference scale for the arrow is shown at the right top of each panel. The corresponding high pressure system, low pressure system, Africa Easterly Jet, and Tropical Easterly Jet are marked as 'H', 'L', 'AEJ' and 'TEJ', respectively.**

## 3.2 Hydroclimatic responses to the lakes in NAf

We investigated the influence of lake distribution in NAf on the hydroclimatic response by analyzing the differences
between our lake sensitivity simulations and the MH$_{ref}$ for summer. First, we examined the influence of the presence of
Western Sahara lakes and Megalake Chad. Without the Western Sahara lakes (MH$_C$), Megalake Chad marginally changed
the local precipitation and water transportation (shaded areas and vectors in Fig. 3a). However, owing to the western lakes
(MH$_{WC}$), the precipitation belt (originally at ~10° N in Fig. 3a) strengthened, expanding northward and eastward to Megalake
Chad (shaded areas in Fig. 3b), and was associated with the enhanced anticlockwise water vapor transportation (vectors in
Fig. 3b). These findings suggested that the Western Sahara lakes enhanced the northward WAM extension. We further
compared the MH$_{WCE2}$ and MH$_{WCE4}$ experiments (Fig. 3c and 3d) to MH$_{ref}$ to assess the impact of the size of Megalake Chad
on the hydroclimatic influence of western lakes. We found that the western lakes at 10°–20° N induced an enhanced
precipitation belt with northwestward water transportation in the MH$_{WCE2}$ experiments (Fig. 3c). With the expansion of
Megalake Chad and eastern lakes, the precipitation belt extended eastward with a strengthened positive response (Fig. 3d),
suggesting the influence of Megalake Chad in eastward monsoon extension.

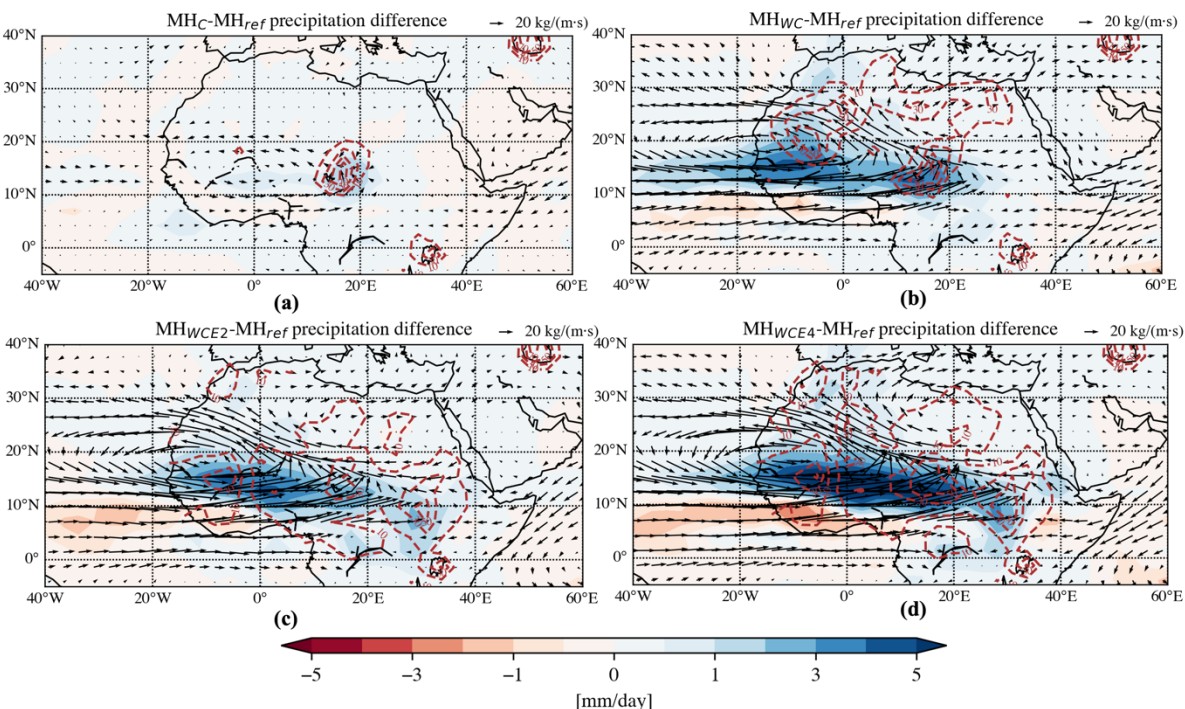

**Figure 3: Anomalies relative to MH$_{ref}$ in the simulated MH climatological summer mean (JJAS) precipitation (shades) and integrated vapor transportation (arrows) for (a) MH$_C$, (b) MH$_{WC}$, and (c) MH$_{WCE2}$ and MH$_{WCE4}$, respectively. For (a)–(d), the lake fraction (%) contours of the respective lake sensitivity experiment are shown as red dashed lines (contour spacing: 10%, 30%,**
**50%, 70%, 100%), and the respective reference scale for the arrow is shown at the right top of each panel.**

To further investigate the mechanisms of the monsoon response to lake expansions, we analyzed the responses in land surface climate variables (soil moisture (SM), evaporation (Evap), and surface temperature (T2); shaded areas in Fig. 4) and atmospheric circulations (geopotential height and horizontal winds; contours and vectors in Fig. 4). In $MH_C$, Megalake Chad

did not affect atmospheric circulations, but it affected the local hydrological cycle with slight increases in SM and Evap by 0.2 m and 2 mm/d, resulting in surface cooling around Megalake Chad by –0.4°C (Fig. 4a, 4b, and 4c). In $MH_{WC}$, the western lakes induced similar local responses, with increased SM and evaporation flux accompanied by a surface cooling in northwest NAf, but with a stronger response than that around Megalake Chad (Fig. 4d, 4e, and 4f). The expansion of the western lakes also impacts the atmospheric circulation. In the upper troposphere (200 hPa), TEJ was enhanced at 5°–15° N (vectors in Fig. 4d). Furthermore, the anticlockwise anomalies of horizontal winds in the middle atmosphere (vectors in Fig.

4e), associated with the weakened Sahara High (contours in Fig. 4e), suggested that the AEJ weakened and shifted northward. In the lower atmosphere, the enhanced monsoon westerly flow at around 10°–20° N (vectors in Fig. 4f) was associated with cyclone circulation over the Atlantic Ocean at approximately 20°–30° N, next to the weakened SHL (contours in Fig. 4f).

Similar responses on the hydroclimatic variables and atmospheric circulation of $MH_{WC}$ were also found in $MH_{WCE2}$ and

$MH_{WCE4}$. The increases in SM, Evap, and T2 extended more eastward in $MH_{WCE4}$ (Fig. 4j, 4k, and 4l) compared with those in $MH_{WCE2}$ (shaded areas in Fig. 4g, 4h, and 4i). The associated atmospheric circulation was further enhanced and extended eastward. In particular, the TEJ became stronger, and the Sahara High further weakened with stronger anticyclonic circulation anomalies extending eastward, leading to a weaker AEJ in $MH_{WCE4}$ than in $MH_{WCE2}$ (contours and vectors in Fig. 4g, 4j 4h, and 4k). Moreover, the above cyclonic circulation in the lower atmosphere shifted southeastward at around 20° W,

further extending the monsoon westerly flow eastward in $MH_{WCE4}$ compared with that in $MH_{WCE2}$ (contours and vectors in Fig. 4i and 4l). Notably, owing to the southeastward extension of the cyclonic circulation response, the weak SHL signals in the $MH_{WCE4}$ experiments were counterbalanced and weakened compared with those in both $MH_{WC}$ and $MH_{WCE2}$ experiments (contours in Fig. 4f, 4i, and 4l).

Hence, the enhanced northward WAM forced by lakes can be explained by lake expansions that induce a cyclonic circulation

in the lower atmosphere, accompanied by a weakened AEJ and stronger TEJ associated with weakened Sahara Highs and SHL. Similar mechanisms have been previously identified based on observations and simulations, although their physical mechanisms are still under discussion (Nicholson, 2009; Lavaysse et al., 2010; Klein et al., 2015; Nicholson and Klotter, 2020). Furthermore, we found that the lake-induced precipitation and SM increment were close to those induced by orbital forcing only, but were restricted over ~10° N (Fig. S5a and S5b). This confirms that lake expansion considerably affected the

humidification of NAf. In summary, Western Sahara lakes and Megalake Chad could enhance the northward WAM triggered by orbital forcings, resulting in a significant humidifying effect.

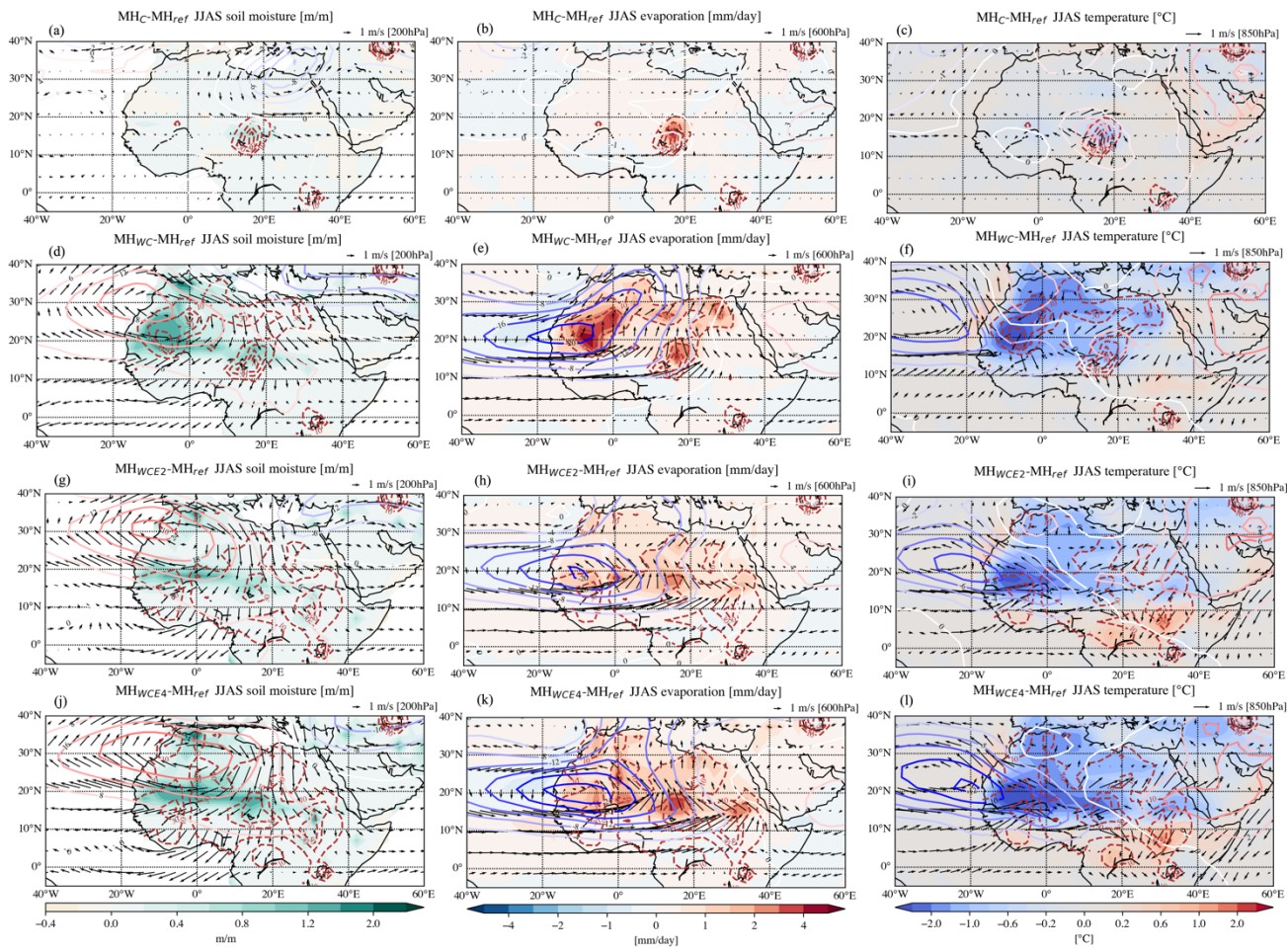

Figure 4: Simulated MH climatological JJAS mean anomalies with respect to $MH_{ref}$: (a) soil moisture (shades) with 200 hPa wind (arrows) and geopotential height (contours), (b) evaporation (shades) with 600 hPa horizontal wind and geopotential height, and (c) surface temperature (shades) with 850 hPa horizontal wind and geopotential height. Maps (d), (g), and (f) are the same as (a), (b) and (c), respectively, but for the $MH_{WC}$ experiment. Maps (g), (h) and (i) are the same as (a), (b), and (c), respectively, but for the $MH_{WCE2}$ experiment. Maps (j), (k), and (l) are the same as (a), (b) and (c), respectively, but for the $MH_{WCE4}$ experiments. For all the maps, the lake fraction (%) contours of the respective lake sensitivity experiments are shown as red dashed lines and the respective reference scale for the arrow is shown at the right top of each panel.

## 3.3 Aridity transformation with lake expansions

To understand the influence of the Western Sahara lakes and Megalake Chad on the hydroclimatic spatial response, we further calculated the anomaly changes of regionally averaged hydroclimate variables with lake expansion over NAf (Fig. 4). Considering $PI_{ref}$ experiments as the reference, the annual mean variables exhibited linear relationships with the mean lake fraction over NAf. The annual mean values of Precipitation (Prcp), Evap, and Net Radiation (Rad) increased with lake fraction, whereas T2 decreased (crosses in Fig. 5). To provide further insights into the changes in Rad, we examined the relationship between net longwave radiation (LW) and net shortwave radiation (SW) in relation to the lake fraction (Fig. S6a;

positive downward). Taking the MH*WCE4* experiments as an example, our analysis revealed that the increase in Rad can be attributed to two factors: increase in downward LW in the cooling and humidifying areas (Fig. S6b) and slight increase in downward SW in the regions with higher lake fraction, which is associated with changes in surface albedo (Fig. S6c). These

findings suggest that the humidifying and cooling areas experienced greater incoming LW radiation absorption.

Additionally, seasonal analysis showed that during summer, the lake sensitivity experiments and the PI$_{ref}$ had considerable differences, with positive anomaly offsets for Prcp, Evap, and Rad and negative anomaly offsets for T2 (upward triangles in Fig. 5). In contrast, during winter, these variables were not significantly related to the lake expansion (standard deviation is approximately 0.1), but a cooling effect was still observed (downward green triangles in Fig. 5). Therefore, the lake

expansion mainly affected hydrological changes in summer, leading to wetter and cooler conditions in the lake sensitivity experiments compared with MH$_{ref}$. However, the unusually high anomalies observed during summer in the MH$_{WC}$ experiments suggest that the position of the lake may play a more important role than the proportion of lakes in moistening the Sahara regions.

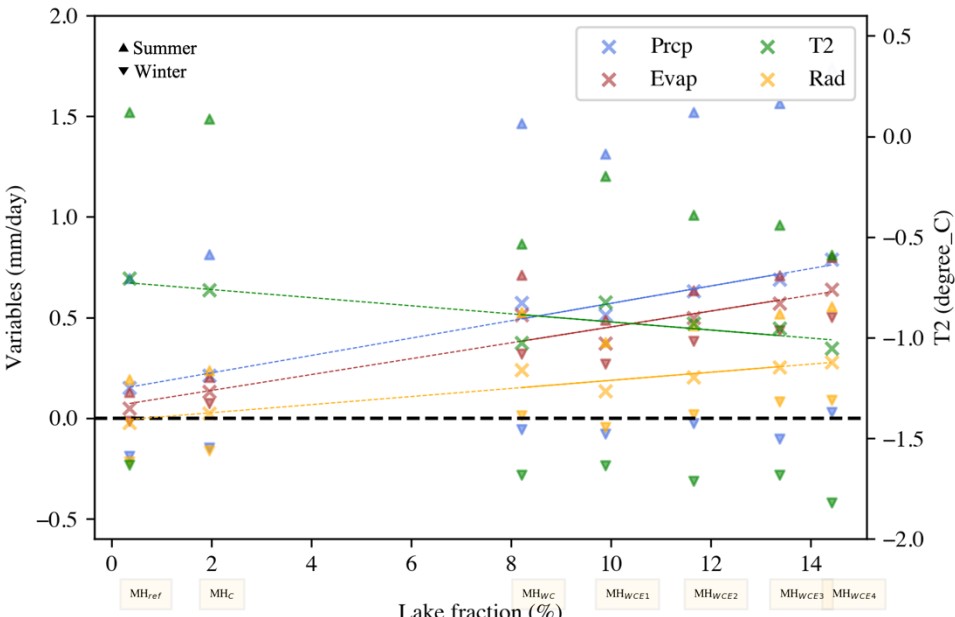

**Figure 5: Statistical relationship between regionally averaged hydroclimate variables anomaly and grid lake fraction over Northern Africa (20°W–40°E, 0–35°N) for MH lake experiment anomalies (relative to PI$_{ref}$) on the annual (cross), JJAS (upward triangle), and JAM (downward triangle) averages. The hydroclimatic variables include precipitation (Prcp [mm/day]), evaporation (Evap [mm/day]), 2 m air temperature (T2 [°C]), and radiation (Rad [mm/day], downward as positive). The p-value is <0.05 for all relationships.**

We used the Budyko aridity index to detect changes in hydroclimatic conditions related to lake expansion. Compared with the MH$_{ref}$ experiments (Fig. S7a), the northwest climate zones changed from hyper-arid to arid and semi-arid zones due to the lake expansions in our six MH sensitivity experiments. Moreover, the western arid and semi-arid zone areas reduced with

increasing northward humid and semi-humid zones, along with increasing tropical humid zones (Fig. S8). Additionally, in the MH$_{WCE4}$ experiments, such climate zones extend further eastward, corresponding to the spatial response of hydroclimatic

variables. Correspondingly, the mean Budyko aridity index anomaly over NAf relative to the PI$_{ref}$ increased with lake expansion, indicating that the aridity extent was lower with the presence of lakes (dots in Fig. 6a). The climate zone transformation indicates the essential role of lake-climate impact in sustaining the northwest humidification of the Sahara by changing the hydroclimatic conditions and alleviating aridity.

However, our climate zone results show that hyper-arid and arid zones remain over northwestern Sahara. Therefore, we

further demarcated regions of the precipitation scarcity and surplus based on the threshold of semi-humid climate zones (I = 2). By comparing the simulated precipitation with the semi-humid climate zone threshold, the regions that receive less than the threshold are considered scarce and regions receiving more are surplus. The total amount of precipitation scarcity was approximately 140–160 mm/d, and the precipitation surplus was approximately 260–370 mm/d over Naf, continuing to increase with lake expansion (bars in Fig. 6a). Compared with the MH$_{ref}$ results, the MH$_{WCE4}$ experiments potentially reduced

precipitation scarcity by up to 13% and increased precipitation surplus by approximately 40%. The spatial patterns showed that the north-dry and south-wet precipitation pattern (Fig. S7b and S9) and the dividing line moved up to about 5° to the north compared with the MH$_C$ experiments over the western NAf regions (Fig. 6b). Additionally, precipitation scarcity values were lower in the western region and higher in the eastern region.

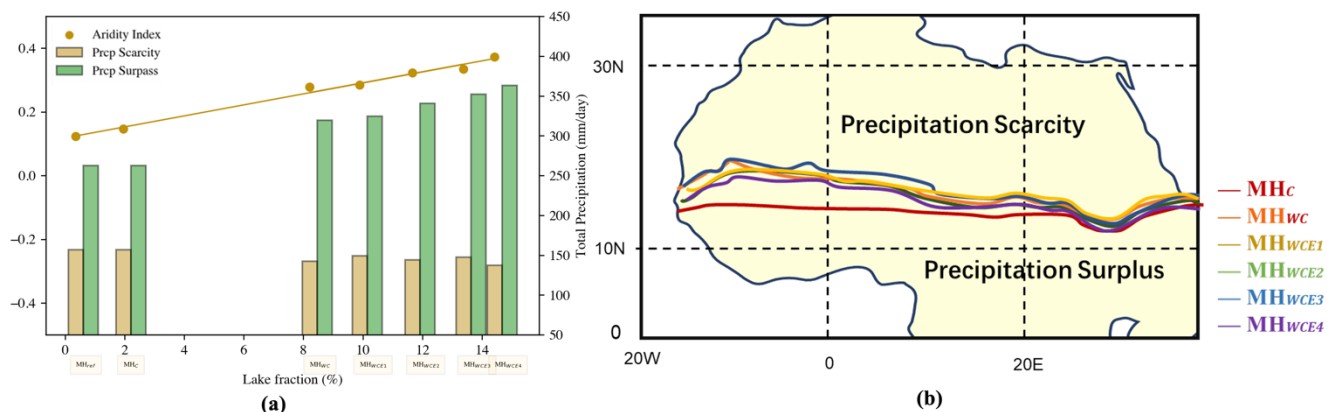

**(a)**                                                                                                      **(b)**


**Figure 6: (a) Budyko Aridity index anomaly between PI$_{ref}$ and MH simulations (left y-axis; unitless) with different averaged grid lake fractions and total precipitation scarcity and surplus amounts (brown and green bars, respectively; mm/d) corresponding to the right y-axis. All the variables are climatological mean annual values. (b) Border between regions of precipitation scarcity zones and precipitation surplus zones for all the MH experiments.**

Notably, such north–south inverse patterns were also observed in the spatial responses of SM (Fig. 4g and 4j), Evap (Fig. 4h and 4k), and T2 (Figs 4i and 4l). In particular, SM and Evap showed positive anomalies with a cooling effect in the north of 10° N, and minor or negative anomalies with a warming effect in the south of 10° N over NAf. However, such near-equatorial (around 0°–10° N) warming effect cannot be explained solely by the reduced precipitation in MH$_{WCE2}$ and

MH$_{WCE4}$ as the enhanced precipitation belt covered the entire tropical area (0°–20° N), in contrast to being concentrated in the WAM regions (around 10°–20° N) in MH$_{WC}$. To identify the inverse temperature anomaly patterns in MH$_{WCE2}$ and MH$_{WCE4}$, we analyzed the stable oxygen isotope ratio ($\delta^{18}$O) in precipitation (Fig. S10). Positive $\delta^{18}$O anomalies suggested the presence of an oceanic moisture source in addition to the local lakes, whereas negative anomalies indicated the influence of local water cycling. The $\delta^{18}$O increase in the northern regions suggests that the moisture sources from the Atlantic Ocean were associated with westerly monsoon winds. Conversely, the equatorial land areas show decreases in $\delta^{18}$O, which indicate weakened evaporation (Fig. 4k) and warming effects (Fig. 4l). Further examination of the $\delta^{18}$O decrease (Fig. S10d) in the equatorial land areas suggested that the slight precipitation increment (Fig. 4d) was not driven by the westerly monsoon winds. Instead, this warming effect induced by equatorial lakes may link to the differences in lake heating during daytime and nighttime (Thiery et al., 2015). Hence, although lakes in WAM regions tend to result in wetter and cooler climatic responses, lakes located elsewhere (such as the eastern lakes in South Sudan) may not impact the northward WAM movement.

## 4 Discussion and Conclusions

We used the MIROC5-iso model with different GS lake maps to investigate the influence of the Western Sahara lakes and Megalake Chad on the northward movement and eastward expansion of WAM, which causes the humidity in the Sahara region. Our results showed that Western Sahara lakes promote the northward movement of WAM, and Megalake Chad can further enhance the monsoon westerly flow response eastward. This cyclonic response in the lower atmosphere is associated with weakened AEJ, SHL, Sahara Highs, and strengthened TEJ (Fig. 7). Additionally, the humidifying transformation of the climate zone and reduction in precipitation scarcity over NAf further highlighted the significant influence of lake expansion in reconstructing the GS climate.

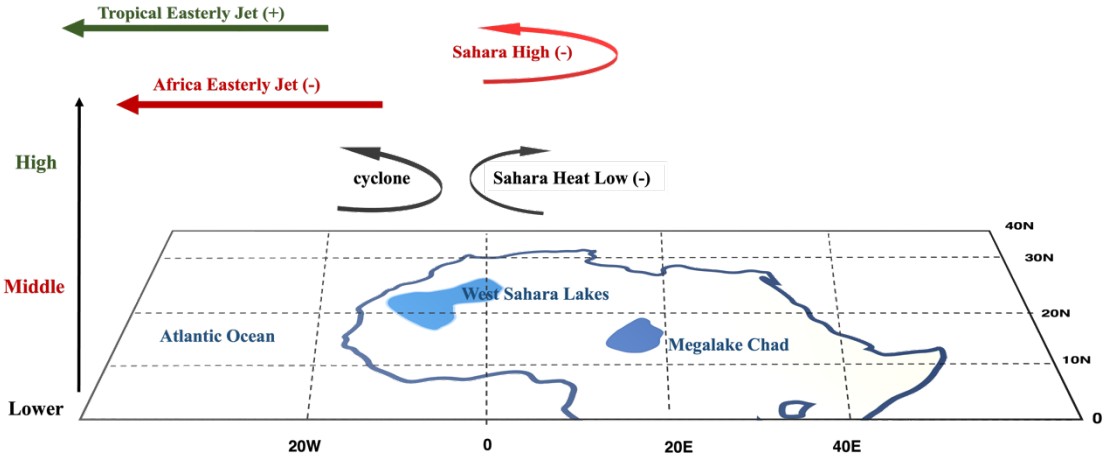

**Figure 7: Lake-climate impact mechanism over North Africa in the MH. The lower, middle, and high atmosphere circulations are marked with black, red, and green colors, respectively. The weakening signal is represented by '−', and the strengthening signal is represented by '+'.**

Our study confirmed that Megalake Chad does not influence the northward monsoon movement without the Western Sahara lakes (Broström et al., 1998; Carrington et al., 2001; Chandan and Peltier, 2020). We also confirmed the influence of Western Sahara lakes on the northward monsoon movement (Specht et al., 2022), but further stressed that Megalake Chad could extend the westerly monsoon eastward when accompanied by Western Sahara lakes. Moreover, compared with our simulations (Fig. S11), Chandan and Peltier (2020) underestimated the contribution of lakes, which were close to the $MH_{WC}$

experiment results, by assuming that the weakened SHL induced by the surface cooling effect would reduce precipitation. However, we found that such an SHL weakening effect can be offset by the adjacent cyclonic circulation response in the lower atmosphere, which promotes precipitation. Moreover, we found that the northward WAM movement corresponded with a weakened AEJ and strengthened TEJ, which is in agreement with Specht et al. (2022). Therefore, we emphasized the importance of how the climate model represents the AEJ and TEJ behaviors to reproduce the MH climate (Claussen et al.,

2017; Bercos-Hickey et al., 2020; Ngoungue et al., 2021).

Furthermore, in terms of the lake position (Chandan and Peltier, 2020; Specht et al., 2022), both the western lakes and Megalake Chad located in the WAM regions could have played a crucial role in inducing the monsoon movement. Finally, the influence of Sahara lakes on the climatic zone transformation in NAf is highlighted, as corroborated by the Budyko aridity index. Such lake-climate response can humidify the GS by transforming the climate zones from hyper-arid or arid to

semi-arid or semi-humid, especially over the northwestern areas, and reduce the precipitation scarcity by up to 13%. However, our lake sensitivity experiments may not comprehensively capture the impact of small lake aggregates, which may limit the scope of our findings. In this study, we included the precipitation and isotope anomalies (Fig. S12), as well as the SM, Evap, and T2 with the low-mid-high level circulation responses (Fig. S13) for $MH_{WCE1}$ and $MH_{WCE3}$. The similarity of these results with $MH_{WCE2}$ and $MH_{WCE}$ confirms that the small lake aggregate effect is negligible in the large-scale lake-

climate impact mechanisms. Nonetheless, conducting ideal sensitivity experiments in the future is necessary to confirm our findings and fully elucidate the impact of lakes on the regional hydroclimate during the MH period.

Limited by the model integration and uncertainty, especially the lack of the dynamic lake or vegetation modules coupled with MIROC5-iso, the model-dependent findings of this study only focused on how the changes in the presence of lakes in terms of surface boundary conditions influence the GS hydroclimatic conditions; it did not consider the climate

reinforcement on lake expansion or shrinkage. Additionally, under the forcing of lake presence, the soil properties and vegetation growth changes also influence the water holding capacity, which determines the greening process. However, these changes are limited by the simplified single-direction impact discussion. Furthermore, due to the absence of coupling with the ocean GCM, the model failed to consider the interactive effects of lake and sea surface temperature or sea ice concentration, which are crucial for the examination of the teleconnection between the ocean and WAM. Hence, dynamic

model integration is required to provide new insights and understand single variable interactions and their joint effect on

land–atmosphere interaction during the GS period (Dallmeyer et al., 2020). Moreover, understanding external forcing, such as orbital parameters and greenhouse gas changes, which influence the GS climate system, would also provide insights into replicating the GS climate in the future (Duque-Villegas et al., 2022). Thus far, the interactive dynamic understanding among potential GS climate drivers remain unclear; different types of interactive feedback mechanisms contributing to the

limitation of the uncertainty should be identified through climate proxy datasets.

In summary, our study identified lake expansions during the MH that sustain the Sahara greening with a northward movement and eastward extension of the WAM. Limited by model dependency, particularly the inclusion or exclusion of certain feedback mechanisms such as dynamic lakes and vegetation modules, and differences in model components and parameterizations used in different studies, the land–atmosphere interaction mechanism forced by dynamic lake changes

remains unclear. Additionally, while the main features of the WAM have been adequately captured, higher resolution simulations are required to simulate finer convective activities and provide new insights at the sub-grid scale (Steinig et al. 2018; Ohgaito, et al., 2021). In the future, the dynamic lake module will be improved to detect the lake-climate interaction with time-varying lake extent in the simulations. Such research will reveal the dynamic interactive mechanism of lake-climate interactions and possible conditions sustaining the Sahara greening processes.

## Code Availability

The code of the isotopic version MIROC5-iso is available upon request on the IIS's GitLab repository http://isotope.iis.u-tokyo.ac.jp:8000/gitlab/miroc-iso/miroc5-iso (Okazaki and Yoshimura, 2019).

## Data Availability

The paleo small lake reconstruction maps (Hoelzmann et al., 1998) and potential maximum lake reconstruction maps (Tegen et al., 2002) used in this study for comparison are the processed ones published by Specht et al. (2022), available at http://hdl.handle.net/21.11116/0000-0009-63B5-B. The updated 15 arc-second lake maps over the NA (Chen et al., 2021) are available at Mendeley Data (http://dx.doi.org/10.17632/8vfhhv8s2f.1); we used the RFM2 model results in this study. Isotopic proxy datasets from ice cores used for the climate model validation method are reported in Table 1 of Cauquoin et al. (2019). The SISALv2 dataset is available at https://doi.org/10.17864/1947.256 (Comas-Bru et al. 2020).

## Author contributions

KK and OT designed the research idea. YL and KK contributed to the experiment design. KK and AC provided model code and input data. YL performed the model experiments and results analysis. YL prepared the manuscript with contributions from all co-authors.

## Competing interests.

The authors have no other competing interests to declare.

## Acknowledgments

This work was supported by the Japan Society for the Promotion of Science [KAKENHI; 21H05002], the Environment Research and Technology Development Fund (JPMEERF20202005) of the Environmental Restoration and Conservation Agency of Japan, the Japan Society for the Promotion of Science via Grants-in-Aid 22K21323 and the advanced studies of climate change projection (SENTAN; JPMXD0722680395) from the Ministry of Education, Culture, Sports, Science and Technology (MEXT), Japan.

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
