# Peer review of "Contribution of lakes in sustaining the Sahara greening during the mid-Holocene"

_Climate of the Past, 2023_

## Referee Comment (RC2)

**Review of: Li et al, 2023, Climate of the Past**

April 26, 2023

**Summary**

The authors present an interesting study for the effect of mid-Holocene lakes. I believe this study and several other studies are still needed to understand the effect of those lakes on the climate of mid-Holocene Africa. As it currently stands, however, the manuscript needs considerable reworking before it is able to make a useful contribution to the compendium of literature on this topic.

**Major comments**

1. **Model choice and experiment setup:** The employed model resolution is T42 (280km) which is very very coarse by today's standards. Some climatic features do depend in a noticeable way on the model resolution. This therefore leaves a lot of questions, in my mind about the underlying results. Furthermore, the experiments have been spun-up for only 30 years and the results have been averaged over another 30 years, which is also not great. Considering the low resolution of the model, it should be possible to integrate it for a longer period of time.

There are two further issues with the model, first being that it does not appear that the model has been run in a fully-coupled model (I am inferring this because it is not explicitly stated and because there are comments about initialization of ocean surface variables, but correct me if I am wrong). This leaves out important interactions with the ocean. Secondly, the SST, sea ice concentration, and the sea surface water isotope distribution are taken from an entirely different model. All these facts taken together present a very unsatisfactory picture of the experimental setup. I think the authors should revise their setup, or, provide sufficient evidence that their setup is not creating adverse results.

2. **Methodology for analysis:** The authors investigate the contribution of the Western Sahara lakes by comparing the $MH_C$ and $MH_{WC}$ experiments, while the effect of Megalake Chad is studied by contrasting the differences between $MH_{WCE2}$ and $MH_{WCE4}$. I do not believe this is the right way of doing sensitivity studies for the effect of either of these two feature; this is because none of the lake maps employed in these simulations differ strictly with regards to those two features. There are several other differences between the lake maps that are all over the place. To some those differences very well may look small enough to ignore, but they don't look small to me (especially considering their aggregate effect over the entire North Africa) and the authors have not provided any evidence supporting their choice to overlook those differences. Rather than comparing $MH_{WCE2}$ and $MH_{WCE4}$ to study the effect of Megalake Chad, a more appropriate thing to do would be to compare

the results from (let's say) MH$_{WCE2}$ with another simulation in which only the employed surface map is the same one as that in MH$_{WCE2}$ but with Megalake Chad removed. Similarly for studying the effect of western lakes (in this case the underlying lake maps MH_98 and MH_02 have lot of other differences over the northern parts of North Africa, Figure S2 of the manuscript).

**3. Isotope feature:** I do not follow how the isotope feature of the model is contributing to this version of the manuscript. The only real result discussed is the global-scale comparison with proxy derived isotope records, but the usefulness of that is lost on me as the subject of the paper is Africa/North Africa and there is only one $\delta^{18}O$ proxy in all of Africa. It is in no way contributing to the understanding of the effect of mid-Holocene lakes over North Africa.

**4. Comparison to proxies:** In contrast to the single $\delta^{18}O$ proxy in all of Africa, there are decent compilations of temperature and precipitation proxies over mid-Holocene Africa [Bartlein et al., 2010] that have been used for validation purposes in many studies. Why are the simulated temperature and precipitation not compared to those proxies?

**Other comments**

**5.** Line 45: Chandan and Peltier [2020] did not use the 'small-lake map' of Hoelzmann et al. [1998]. The Hoelzmann map prescribes a small uniform lake fraction for nearly all of Sahara, this aspect was not utilized in their paper. Furthermore, the Hoelzmann map includes a sizeable region of wetlands covering >70% grid cell south-east of Megalake Chad which is not included in the Chandan and Peltier land surface. Actually, on this matter, I wonder why these wetlands are not included in your Hoelzmann map considering that you say in the manuscript that you treat wetlands as lakes? I am also curious why your Hoelzmann map differs noticeably from what is shown in Plate 3 of Hoelzmann et al. [1998]?

**6.** Please revise/rewrite the content between lines 60 and 66. It is not quite clear what discrepancies you are trying to highlight in these lines.

**7.** There are too many names in the paper that start with MH and which refer to both simulations and lake maps. This makes reading the paper rather confusing as I easily mix up lake map names with experiment names. I suggest keeping the experiment names as they are and renaming the lake maps to LK (or something else). For example, MH_98 lake map becomes LK_98.

**8.** Provide more information on the Budyko aridity index in section 2.3.2, including but not necessarily limited to how it should be interpreted, what is the physical basis for this metric

and what are the caveats of using this metric.

**9.** Section 3.3 is very difficult to follow. I suggest a complete re-write of this section. Here are some of my comments for that section.

- Line 239: What radiation is this? Longwave downwelling? Why does it increase with lake fraction?

- I do not follow lines 240–246.

- The text says that Figure 4 shows zonally averaged quantities but that is clearly not the case. What averaging is being done in Fig 4?

- Fig 4 caption: how can the units of radiation be "mm/day"? Where is the vertical axis for radiation data?

- What do 'precipitation scarcity' and 'precipitation surplus' mean? Scarcity and surplus with respect to what? Please define them clearly. How are figures S5b and S7 showing these quantities generated? How are the numbers presented in line 165 and shown in Fig 5a computed? I cannot make sense of these results because you haven't defined the two phrases.

- Line 264 "implying that ... wetter." this remark does not make sense when read within the full sentence.

- Line 267 ""The spatial pattern showed .... modes." What mode? I don't see any (dynamical) mode here, it is just the northward extent of the WAM which starts from the south. Did you mean to say a 'precipitation pattern'?

- Line 279 What is this inverse pattern?

- Line 280 There is nothing new in the finding that the moisture source is largely oceanic along with some contribution from local moisture recycling. Is the isotope analysis contributing anything new?

- Line 283 What inverse warming effect?

**10.** Line 302–303: I am not sure it is correct to say that Chandan and Peltier [2020] underestimated the contribution of lakes (similar sentiment regarding Line 47). In their study, the lakes do have quite a bit of contribution in the 10–15N latitudinal band which is the same region where precipitation effect is greatest in your simulations. If you look at Figure 3 of that paper, the influence of lakes, determined by the zonal mean difference between MHV and MHVL, can be as high as 200mm or more in that latitudinal band, and while a spatial difference between those two simulations was not shown in that paper, I am quite sure it would be very similar to the spatial patterns shown in your Figure 2. Are you able to compute an equivalent zonal precipitation mean to compare with CP2020's Fig 3 and thus argue that the lake influence in their lake experiment is decidedly lower than in yours?

**11.** Line 309: "we suggest that western lakes and Megalake Chad should be located in the WAM regions to induce the monsoon movement" I am not sure what you mean by that. One

doesn't get to choose where any lake is located, it is located where it is (or was).

12. Figure S2: For sub-figure (g), how is the lake fraction defined? Is it *lake_area_africa/area_global*? Or is it *lake_area_global/area_global*? Why not just use *lake_area_africa/area_africa*? I don't see the need for anything 'global' in calculating lake fractions as everywhere outside of North Africa the lake map is unchanged. Furthermore, lake fraction in terms of the area of Africa (say north of equator) yields a number that can be better compared to other numbers in the literature. Please also address the sentence on lines 109–110 based on your revision.

13. Figure S5: The description for sub-figure (b) is wrong.

**Technical comments**

The paper could use a through examination for grammar and clarity. Here are just some selected instances, but there are several more that I didn't have the time to put here.

14. **Line 81:** the hydroclimatic influence of  the presence of lakes

15. **Line 82:** two control simulations  for the

16. **Line 90:** sea surface provided by MPI-ESM-wiso  (Cauquoin et al., 2019) as boundary conditions for our PI and MH simulations

17. **Line 92–93:** It doesn't make sense to say you "found few lakes existed in NAf", because of course very few lakes exist in the NAf today. Please re-phrase.

18. **Line 91–94:** Please move the remark starting on this line (i.e starting from 'Figure S1a shows...') immediately before the sentence on line 87 which starts with 'Each experiment was run.'

19. **Line 102:** MH_98 lake map .... with only  Megalake Chad

20. **Line 107–108:** Please rephrase the line "MH4 accounting...."

21. **Line 113:**  Megalake Chad's influence on NAF climate  was assessed using  ......

22. **Line 120:** These are  presented in Table 1

23. **Line 122:** which are reported in  Risi et al 2010.

24. **Line 134:** component of the vertically integrated

25. **Line 136:** where u is the  zonal wind

26. **Line 137:** The meridional component of the vertically integrated

27. **Line 154:** "verified based on" or "verified in"

28. **Line 155:** of global MH  characteristics using the MIROC-series

29. **Line 196:** What is SM? Soil Moisture?

30. **Line 263:** we further  demarcated regions of precipitation...

31. **Line 274:** The  border between regions of precipitation scarcity  and precipitation surplus ...

32. **Line 288:** Difficult to follow. Please re-write this sentence.

**References**

Patrick J Bartlein, Sandy P Harrison, S Brewer, S Connor, B A S Davis, K Gajewski, J Guiot, T I Harrison-Prentice, A Henderson, O Peyron, Ian Collin Prentice, M Scholze, H Seppä, B Shuman, S Sugita, R S Thompson, A E Viau, J Williams, and Hanbo Wu. Pollen-based continental climate reconstructions at 6 and 21 ka: a global synthesis. Climate Dynamics, 37(3-4):775 – 802, 09 2010. doi: 10.1007/s00382-010-0904-1.

Deepak Chandan and W. Richard Peltier. African Humid Period Precipitation Sustained by Robust Vegetation, Soil, and Lake Feedbacks. Geophysical Research Letters, 47(21), 2020. ISSN 0094-8276. doi: 10.1029/2020gl088728.

Philipp Hoelzmann, D Jolly, Sandy P Harrison, F Laarif, R Bonnefille, and H. J. Pachur. Mid-Holocene land-surface conditions in norther Africa and the Arabian peninsula: A data set for the analysis of biogeophysical feedbacks in the climate system. Global Biogeochemical Cycles, 12(1):35–51, 1998.

---

## Author Comment (AC1)

**Responses to Reviewer #1's comments:**

**Reviewer #1 General comments:** *This paper adds to a body of studies of the effect of lakes during the North African "Green Sahara" mid-Holocene period. As it is rightly stated, there is still a discussion about which processes have enabled and sustained a relatively humid climate in that region during that period, and besides in particular vegetation, open water is one surface feature that has been proposed as a positive feedback mechanism involved in this interesting period of « recent » climate history. The manuscript does not add fundamentally new insights to this discussion, but as it stands, it is a basis for a useful contribution to this discussion, provided some necessary clarifications. These clarifications are needed in particular with respect to the model setup.*

**A:** We thank the reviewer for his/her general appreciation of our paper and for the constructive comments and corrections that helped to significantly improve this manuscript. We have carefully revised it as described in detail below. We would like to acknowledge that we have made corrections to figures 1-4 and figures S4 and S8 to address a mistake in the seasonal calculation. Specifically, some of the previous results displayed the May-Oct mean instead of the Jun-Sep results. This initial discrepancy has no impact on our overall findings. For the corrections in the manuscript, we provide the line numbers from the revised paper with track changes.

**Reviewer #1 Comment 1:** (hereafter referred to as R1C1, R1C2…) *Line 20: Editorial - In several places in the introduction, reference is made to a recent review paper instead of older key papers. For example here in line 20 where only a (good and complete) review is cited, it might be interesting to expand the list of papers cited to include some preceding key papers. However, that's an editorial question and it is also acceptable to only cite the review paper, for clarity.*

**A:** Thanks for your suggestion. As you suggested, more references have been

supplemented in the introduction parts to strengthen the reasoning.

Line 20-21: The references have been revised as "(Gasse, 2000; Adkins, deMenocal, & Eshel, 2006; Claussen, M. et al., 2017)."

Adkins, J., deMenocal, P., & Eshel, G. (2006). The "African humid period" and the record of marine upwelling from excess 230Th in Ocean Drilling Program Hole 658C. *Paleoceanography, 21*(4). doi:https://doi.org/10.1029/2005PA001200

Gasse, F. (2000). Hydrological changes in the African tropics since the Last Glacial Maximum. *Quaternary Science Reviews, 19*(1), 189-211. doi:https://doi.org/10.1016/S0277-3791(99)00061-X

**R1C2:** *Line 68: A general remark: This paper used an isotope-enabled version of a GCM. I expected some more isotope-related analyses in this paper, for example to provide insights into precipitation recycling in the various simulations. I was a bit frustrated not to see more on this, as this might add some rather unique information from this study.*

**A:** Thank you for your comment and for bringing attention to the importance of isotope-related analyses in our study. We agree that such analyses can provide unique insights into the water cycle dynamics simulated by the isotope-enabled GCM, and we would like to clarify that the use of an isotope-enabled model was primarily aimed at capturing these dynamics, rather than solely for model-data comparison purposes.

To address this point, we have made additional clarifications in both the Method and Result analysis. In section 2.1 Lines 102-104: "Such isotope-enabled climate models have proven to be valuable tools for tracing water vapor transportation and identifying the sources of precipitation changes (Tharammal, T. et al., 2021; Liu, X. et al., 2022)."

In the Result section, we further analyzed the stable oxygen isotope ratio in precipitation to differentiate the source of increasing precipitation from ocean and land. We also made additional revisions in section 3.3 Lines 426-435: "Positive $\delta^{18}$O anomalies suggested the presence of an oceanic moisture source in addition to the local lakes, whereas negative anomalies indicated the influence of local water cycling. The $\delta^{18}$O increase in the northern regions (Figure S10) suggests the moisture sources from the Atlantic Ocean are associated with westerly monsoon winds. Conversely, the equatorial land areas show decreases in $\delta^{18}$O, which are also current with weakened evaporation (Figure 3k) and warming effects (Figure 3l) in MH$_{WCE4}$. Further examination of the $\delta^{18}$O decrease (Figure S10d) in the equatorial land areas in MH$_{WCE4}$ suggested that the slight precipitation increment (Figure 2d) was not driven by the westerly monsoon winds. Instead, such a warming effect induced by equatorial lakes may link to the differences in lake heating during daytime and night (Thiery et al., 2015). Hence, while lakes in WAM regions tend to result in wetter and cooler climatic responses, lakes located elsewhere (such as the eastern lakes in South Sudan) may not impact the northward WAM movement."

The use of isotopic features in the model allows us to validate our simulations against paleo-proxy records, avoiding bias from reconstructed datasets. However, these revisions emphasize that our use of an isotope-enabled model goes beyond model-data comparison and provides valuable insights into the water cycle dynamics and precipitation recycling processes in the region under study.

Tharammal, T., Bala, G., Paul, A., Noone, D., Contreras-Rosales, A., & Thirumalai, K. (2021). Orbitally driven evolution of Asian monsoon and stable water isotope ratios during the Holocene: Isotope-enabled climate model simulations and proxy data comparisons. Quaternary Science Reviews,252, 106743.

Liu, X., Xie, X., Guo, Z., Yin, Z. Y., & Chen, G. (2022). Model-based distinct characteristics and mechanisms of orbital-scale precipitation δ18O variations in Asian monsoon and arid regions during late Quaternary. National Science Review.

**R1C3:** *Line 79: Nowadays, T42 is on the lower end of usual climate model resolutions. Is there a reason to think that the results might be sensitive to resolution? For example, are there higher-resolution studies of the West African Monsoon system with MIROC, and is the monsoon representation in MIROC sensitive to model resolution?*

**A:** Thank you for your comments on the issue of model resolution. As you noted, T42 is indeed on the lower end of usual climate model resolutions. In our study, T85 simulation is our another choice but due to the number of sensitivity experiments and computational constraints, we finally chose to use the T42 resolution simulation with the isotope-enabled MIROC5-iso.

As for the high-resolution simulation on West Africa Monsoon (WAM) with MIROC, there seems not so much for the AR5. The latest PMIP4 MIROC-ES2L dataset for 6 ka also has a spatial resolution of T42, indicating the T42 resolution is acceptable for large-scale research (Ohgaito, R. et al., 2021). Besides, Steinig et al. (2018) used the Kiel Climate Model (KCM) to investigate the impact of spatial resolution on WAM precipitation, revealing that higher-resolution models produce similar results to lower-resolution models due to a reduction in convective (subgrid-scale) precipitation and increase in large-scale precipitation. Furthermore, lower resolution models may shift the African Easterly Jet (AEJ) core towards the north and strengthen the Tropical Easterly Jet (TEJ). Thus, the impact of spatial resolution of MIROC on the convective and large-scale precipitation and the position and strength of the AEJ and TEJ, will influence our research findings or not need to be further investigated. However, we agree that it would be interesting to investigate the sensitivity of the monsoon representation in MIROC to model resolution in future research. Hence, we would like to add the model uncertainty in discussion Line 516-518: "Additionally, while the main features of the WAM have been adequately captured, higher-resolution simulations are required to simulate finer convective activities and provide new insights at the subgrid-scale (Steinig, S., et al. 2018; Ohgaito, R. et al., 2021)."

While there is a possibility that our results could be sensitive to model resolution, we believe that our findings are still valid and provide useful insights into the lake influence in Green Sahara. In our studies, we have also performed model validation to ensure that our simulations capture the main features of the West African Monsoon system in section 3.1.

Ohgaito, R., Yamamoto, A., Hajima, T., O'ishi, R., Abe, M., Tatebe, H., ... & Kawamiya, M. (2021). PMIP4 experiments using MIROC-ES2L Earth system model. Geoscientific Model Development, 14(2), 1195-1217.

Steinig, S., Harlaß, J., Park, W. et al. Sahel rainfall strength and onset improvements due to more realistic Atlantic cold tongue development in a climate model. Sci Rep 8, 2569 (2018). https://doi.org/10.1038/s41598-018-20904-1

**R1C4:** *Line 92: "Figure S1a shows..." - Not very clear, figure hard to read. Can the procedure be explained in a bit more detail? I guess the main point is that the lake fraction in MH_ref and PI_ref (note typesetting error line 92, it should be subscript "ref") is weak, right? Because that provides a "almost no lake" reference for the other simulations. Can that be said more clearly?*

**A:** We apologize for any confusion caused by the unclear figure and will make sure to provide a more detailed explanation of the procedure in the revised version of the manuscript.

To address your specific question, the purpose of Figure S1a is to show the spatial distribution of lake fractions in reference simulations ($MH_{ref}$ and $PI_{ref}$). The lake fraction represents the area of the grid cell that is covered by the lake, and in our simulations, we varied the lake fraction in the North Africa (NAf) basin to investigate its impact on the West African Monsoon system.

As you correctly pointed out, the $MH_{ref}$ and $PI_{ref}$ simulations were used as a reference

to represent a scenario with almost no lake in the NAf. In these simulations, the lake fraction was set to a very low value (0.01%). By contrast, in the other simulations, we varied the lake fraction from 0.1% to 1.0%.

In the revised version of the manuscript, we revised in section 2.1 Lines 113-115: "*Figure S1a shows..(Figure S1b).*" to "In $MH_{ref}$ and $PI_{ref}$ experiments, the presence of lakes in North Africa (NAf) is minimal, using the global lake fraction map from the ETOPO5 as in MIROC5 standard simulations (Figure S1). In contrast, the other experiments show highly varied lake fractions, indicating a much higher lake fraction in those cases."

**R1C5:** *Line 110: "1.48 x 108 km$^2$" - please use superscripts correctly. What is the point to compare the lake area over NAf with the global land area? Lake fraction should be relative to the region you are looking at. Or is that the case here? Confusing. If it is relative to the entire land area of the Earth, it's huge...*

**A:** We apologize for the incorrect use of superscripts in the manuscript and corrected them in the revised version.

Regarding the lake area comparison, we agree that the lake fraction should be relative to the region of interest. In our study, we are interested in the lake area changes in the mid-Holocene compared to the present lake, and we have adjusted Figure S2g to reflect this. See section 2.1 at lines 145-147: "The average main lake fraction over the NAf region according to these different reconstructions varies from 1-10 % compared to the total land areas of NAf (Figure S2g)." We have also modified the Figure S2 caption: "(g) The fraction (circle size) of all the prescribed lakes experiments compared to the total land area of North Africa."

Regarding your concern about the large lake areas in LK1-LK4 (Here, we changed the lake map names with 'MH' to 'LK'), this is related to the datasets published in (Chen, Ciais et al., 2021), where potential wetlands (including lake areas) are defined as

persistently saturated or near-saturated areas that are regularly subject to inundation or shallow water tables if there were no human disturbance (Tootchi et al., 2019). Whereas, the LK_98 and LK_02 only include lake maps. We have supplemented more map details in Lines 144-145: "LK4 has the largest lake proportion in the western, eastern, and Megalake Chad regions, and differs from LK2 primarily in its representation of Megalake Chad (Figure S2d, S2f)."; Lines 147-148: "It should be noticed that the water body delineated in LK_98 and LK_02 lake maps only pertain to the lake but the LK1-4 lake maps include both the wetland and lakes.".

**R1C6:** *Line 112: "In this study, wetlands are considered as lakes". What does that mean in the model world? How deep are the lakes? Does that simply mean that the water is present perennially? Please clarify how lakes are prescribed and treated.*

**A: Here, we answer the questions of the reviewer in detail one by one.**

*"In this study, wetlands are considered as lakes". What does that mean in the model world?"*

Regarding the LK_98 and LK_02 maps, we only used the small lake map (Hoelzmann, Jolly et al., 1998) and the maximum lake map (Tegen, Harrison et al., 2002). The details can be found in the data availability and Table S1. However, the latest high-resolution one (Chen, Ciais et al., 2021) includes both the wetland and lakes. Due to our model limitation, the wetland module only accounts for wetland-related processes in middle and high-latitude grids with snowmelt, as described by Nitta et al. (2015, 2017). Hence, these model features were considered in prescribing and treating lakes as wetlands in the MIROC5_iso when simulating the LK1-4 maps.

In order to further clarify this point, we make some revisions on:

Section 1 Lines 77-78: "…… and the recently-updated high-resolution lake and wetland reconstructions maps (Chen et al., 2021) over the NAf during the MH"

Section 2.1 Lines 147-150: "It should be noticed that the water body delineated in LK_98 and LK_02 lake maps only pertain to the lake but the LK1-4 lake maps include both the wetland and lakes. Generally, lakes and wetlands are persistently saturated or near-saturated areas that are regularly subjected to inundation or shallow water tables in the absence of human disturbances (Tootchi et al., 2019). In this study, wetlands are also treated as lakes in our climate model."

*How deep are the lakes? Does that simply mean that the water is present perennially? Please clarify how lakes are prescribed and treated*

The land component of MIROC5-iso is MATSIROC6. The lake module of MATSIRO6 considers lakes as a separate feature, and in this study, only the lake fraction boundary conditions were altered while keeping other boundary conditions constant in control experiments. By default, the maximum lake depth ($H_{max}$=climate + 10m) in MATSIRO6 is set to the climatology of lake depth plus 10 meters, with a minimum depth threshold ($h_{min}$=10m). As the lake depth map was not modified in this study, the lake depth initial values started at the minimum threshold and gradually reached a stable status over time. In our simulated areas, lake depths varied from around 10 to 40 meters in the lake fraction changed areas.

To clarify the description of the lake dynamics in our simulations, we have supplemented the lake module simulation in section 2.1 model introduction Line 98-102: "The MIROC land component is the Minimal Advanced Treatments of Surface Interaction and Runoff (MATSIRO) model (Takata et al. 2003), which could simulate important water and energy circulation. The lake module simulates the thermal and hydrological processes of lakes and their interaction with the atmosphere. It should be noted that a minimum lake depth threshold (10 m) is set, which means the lake permanently existed.".

Takata, K., Emori, S., & Watanabe, T. (2003). Development of the minimal advanced treatments of surface interaction and runoff. Global and planetary Change, 38(1-2), 209-222.

Nitta, T., K. Yoshimura, and A. Abe-Ouchi (2015) A sensitivity study of a simple wetland scheme for improvements in the representation of surface hydrology and decrease of surface air temperature bias. Journal of Japan Society of Civil Engineers, Ser.B1 (Hydraulic Engineering), 71 (4), 955–960.

Nitta, T., K. Yoshimura, and A. Abe-Ouchi (2017) Impact of arctic wetlands on the climate system: Model sensitivity simulations with the MIROC5 AGCM and a Snow-Fed wetland scheme. J. Hydrometeorol., 18 (11), 2923–2936.

**R1C7:** *Line 150: A r^2 of 0.33, is that really good?*

A: An $R^2$ value of 0.33 is indeed quite low. On the other hand, other isotope-enabled model studies for the mid-Holocene period, like Cauquoin et al. (2019) with MPI-ESM-wiso, found $R^2$=0.38 and RMSE=0.79‰. As Cauquoin et al. (2019), we found too low an amplitude of $\delta^{18}O$ changes compared to the observed ones. This is a common bias in isotope-enabled models. Additionally, we observed that around 50% of the data points exhibit positive anomalies in alignment with the observations, while the remaining 50% display negative anomalies. This suggests that our model accurately captures the direction of changes but with a weaker amplitude compared to the observed values. Therefore, we believe that the $R^2$ value of 0.33 with a very low RMSE of 0.81‰ we obtained in our study represents a reasonable correlation between the modeled and observed data, compared to other studies.

**R1C8:** *Line 152: "Our simulations bias..." - Can this be clarified, e.g. by restricting the scatter plot to an area with, say, Africa, Southern Europe and Western Asia?*

**A:** Thank you for your suggestion.

**A:** We agree that restricting the scatter plot to a specific area could provide a more regional perspective on the model biases. Unfortunately, there are few proxy sites only in Africa and West Asia, which limits our ability to constrain the model biases in these regions. However, we would further clarify the simulation bias in North Africa. While

we acknowledge the limited availability of our using proxy records in Africa, the three North African stations for which data is available showed good agreement with the modeled data.

To further examine the model performance in North Africa, we first conducted a comparison with Figure 4a of another study by Larrasoaña et al. (2013). Our findings (Figure R1a) indicate that the MIROC5-iso simulation has difficulty in shifting the zone with precipitation less than 1000 mm/year northward, but it exhibits good agreement with the reconstructed map in the zone with precipitation exceeding 1000 mm/year. This comparison shows the simulation bias of the MIROC5-iso model in North Africa, specifically in terms of the northward movement of the monsoon system.

We also expand the comparison to include another proxy datasets compiled by Bartlein et al. [2010] that would enhance the robustness of our findings. However, we note that the proxy datasets provided by Bartlein et al. [2010] only cover the anomalies between 6ka-0ka, whereas our experiment shows the anomalies between 6ka-PI (1850y). Such difference between 0ka-PI would further bring ignorable bias to our comparison results in addition to the bias from constructed precipitation/temperature datasets. Considering such bias, the comparison results show agreeable changing trends in annual mean precipitation and mean temperature in the warmest month in spatial distribution, but they do not address a good statistical relationship between the proxies and model data (Figure R1 b-e).

In terms of the comparison between precipitation data from our model (Figures R1b and R1c) and the proxy data, we observe good agreement in the central part of North Africa (NAf). However, in the northern region, our model underestimates precipitation compared to the proxy data. These results confirm that our model has limitations in simulating abundant precipitation in the northern region of NAf. Regarding the comparison of summer season temperatures (Figures R1d and R1e), our model generally underestimates temperatures in the central part of NAf but shows good agreement in the northern part. These validation results indicate that our model fails to

capture sufficient precipitation in the northern part of NAf, while precipitation tends to concentrate in the central part with lower temperatures for the mid-Holocene. This discrepancy aligns with the challenge faced by many climate models in reproducing adequate precipitation over NAf. Considering the potential bias introduced by proxy datasets construction and the differences in the study period, we consider the validation results to be acceptable.

This part of the comparison has been added in section 3.1 Lines 249-256: "To further examine the model performance in North Africa, we compare our precipitation result with Figure 4a in the study conducted by Larrasoaña et al. (2013). From Figure S4a, our results indicate that the MIROC5-iso was hard to reproduce the northward shift of the zone with precipitation less than 1000mm/year, but show good agreement with the reconstructed map in the zone with precipitation exceeding 1000mm/year. Besides, we also compared our result with precipitation and summer season temperature anomalies between 6ka-0ka, as provided by Bartlein et al. (2010) (Figure S4b-e). This comparison also revealed precipitation underestimation in the northern NAf and lower temperatures in the central NAf. These comparisons collectively suggest a simulation bias of the MIROC5-iso model in North Africa, particularly concerning the northward movement of the monsoon system."

[Figure]

**Figure R1.** Precipitation and temperature model-data comparison for the reference mid-Holocene simulation in North Africa. (a) The spatial annual precipitation for $MH_{ref}$. (b) shows the simulated global pattern of annual mean precipitation between the $MH_{ref}$ and $PI_{ref}$ climate (background colors) and the observed annual mean precipitation changes (squares) between $MH_{ref}$ and the present climate. (c) is a scatter plot showing a comparison of observed precipitation changes with simulated precipitation anomalies at the same location. (d) and (e) are the same as (c) and (d) but for the seasonal mean temperature model [Summer (JJA)]-data [warmest month] comparison.

Additionally, as described in section 3.1, our model was able to successfully capture

the critical components of the West African Monsoon (WAM), which are particularly relevant to our study of the lake-climate mechanism.

Hence, we acknowledge the importance of regional analyses in future studies when more data become available, and we found that there is simulation bias of the MIROC5-iso model in North Africa regarding the northward precipitation, but the simulation performance in North Africa is acceptable.

Larrasoaña, J. C., Roberts, A. P., & Rohling, E. J. (2013). Dynamics of green Sahara periods and their role in hominin evolution. PloS one, 8(10), e76514.

**R1C9:** *Figure 3: I appreciate that 200, 600 and 850 hPa winds and geopotential heights are also given, but it's unclear whether there is any reason why SM is associated with 200 hPa circulation, evap with 600 hPa, and t2m with 850 hPa. Is there a reason?*

**A:** To clarify, we included 200, 600, and 850 hPa winds and geopotential heights in Figure 3 to provide a comprehensive view of the atmospheric circulation changes associated with the simulated changes in soil moisture, evapotranspiration, and surface temperature. However, there is no specific reason why soil moisture is associated with 200 hPa circulation, evapotranspiration with 600 hPa, and surface temperature with 850 hPa. We apologize for any confusion that may have arisen from our presentation and hope that this clarification helps.

**R1C10:** *Figure 3a: Soil moisture changes. How much of that is prescribed? In the sense, does the prescribed lake water count here? Water quantities are huge, what does 1 m mean here (until what depth?)*

**A:** The soil moisture changes shown in Figure 3a are a combination of both prescribed and modeled changes. The prescribed lake water was not counted towards the soil moisture changes, as the lake water interacts with the surrounding soil and affects its moisture content.

Here, the original soil moisture means total soil moisture $[kg/m^2]$, and the unit was transferred to $[m]$ by dividing by $1000 \ kg/m^3$. Hence the physical meaning is the total soil water column per area or the soil water column per meter [m/m] by dividing 1m depth. We have corrected those units in all of Figure 3.

**R1C11:** *Figure 3c: This is a strong cooling. What is the depth of these lakes? Is is thermal inertia due to depth or evaporative cooling?*

**A:** Thank you for your comments, which raise an important question regarding the cooling mechanism associated with the lake's thermal inertia or evaporation.

The simulated lake depths in North Africa range from 10 m to 40 m. In the lake module, the lake surfaces are considered in the energy balance solution, and each lake layer updates its water temperature based on the incoming downward flux and depth changes, which also is quite important for the lake-climate interaction. However, we assert that evaporative cooling plays a more crucial role based on the simulated results that the spatial distribution of evaporation anomalies and temperature anomalies exhibit similar spatial patterns, as shown in Figure 3. This finding suggests that the evaporative cooling effect may outweigh the influence of lake thermal inertia.

As for the such discussion on the comparison between lake thermal inertia and lake-surface evaporation is quite important for us to understand the lake-climate mechanism, we will do further related research work to understand their roles in energy transmission.

**R1C12:** *Line 269: "Additionally..." - This sentence is not grammatically correct I think.*

**A:** In the revised version, section 3.3 Lines 399-400, it has been revised to: "Additionally, precipitation scarcity values were lower in the western region and higher in the eastern region."

**R1C13:** *Figure 5a: Typo in the legend - should probably be "unitless" (as in the caption), not "uniteless".*

**A:** Thank you for your reminders. We apologize for this spelling error and have modified it in Figure 5a.

**R1C14:** *Line 279: Here are the isotopes, but the explanation is hard to follow for non-specialist readers. This needs and deserves some more explanation.*

**A:** Thank you for your reminders. Please see our response to the R1C2 comment.

This main purpose and findings of the isotope have been answered together in R1C2.

**R1C15:** *Line 314: "Limited by..." - this is confusing, not well written. One wonders whether you have dynamical lakes and vegetation in the model (you don't, if I understand correctly). Please clarify - it would be good to provide a bit more detail in the methods section about this.*

**A:** We apologize for the confusion caused by the wording in "Limited by the model integration and uncertainty, especially the dynamic lake or vegetation modules coupled with MIROC5-iso, ……". We meant that our model can not simulate the vegetation and lake dynamically but treat them as the prescribed boundary conditions for each experiment. Hence, further coupling of MIROC5-iso with dynamic lake or vegetation modules definitely can help us get new insights into the lake/vegetation-climate interaction in future work.

To clarify this in the revised manuscript, we added a sentence in Section 2.1 Lines 110-111: "It should be noticed that the lake fraction is treated as the prescribed boundary conditions in the model based on the corresponding datasets, as the model cannot simulate the lake dynamically." after the explanation of 'Land surface boundary conditions'. Besides, in Section 4 Lines 493-494, we further clarified the original sentence as: "Limited by the model integration and uncertainty, especially the lack of the dynamic lake or vegetation modules coupled with MIROC5-iso, ……"

**R1C16:** *Line 321: "out components, such as orbital forcing and greenhouses..." - you*

*mean "external forcings, such as orbital parameter and greenhouse gas changes" (or something similar)?*

**A:** Thank you for pointing out this untechnical expression. We revise this sentence in section 4 Lines 502-503 to: "Moreover, understanding the external forcing, such as orbital parameters and greenhouse gas changes, …… ".

**R1C17:** *Line 327: "Limited by model dependency and module integration..." - this is unclear. Do you mean to say that the results are highly model-dependent (because the results from different studies are somewhat contradictory), and that they depend on the feedback mechanisms represented (e.g. dynamics lakes and vegetation included or not)?*

**A:** We apologize for the unclear wording. We meant that the results are limited by the model's dependence on certain assumptions and the integration of various modules. These factors can affect the reliability of the results and the understanding of lake/vegetation – climate interaction, especially when it comes to the representation of feedback mechanisms such as dynamic lakes and vegetation.

To make this clearer, we revised it in section 4 Lines 513-516 as: "Limited by model dependency, particularly the inclusion or exclusion of certain feedback mechanisms such as dynamic lakes and vegetation modules, as well as the differences in model components and parameterizations used in different studies, the land-atmosphere interaction mechanism forced by dynamic lake changes remains unclear."

**R1C18:** *Line 331: Full stop missing at the end.*

**A:** Corrected.

**R1C19:** *Supplementary material:*

*Figure S2 - "Experiements" (typo). "(G)" missing in the lowest panel.*

**A:** Corrected.

---

## Author Comment (AC2)

**Responses to Reviewer #2's comments:**

**Reviewer #2 General comments:** *The authors present an interesting study for the effect of mid-Holocene lakes. I believe this study and several other studies are still needed to understand the effect of those lakes on the climate of mid-Holocene Africa. As it currently stands, however, the manuscript needs considerable reworking before it is able to make a useful contribution to the compendium of literature on this topic.*

**A:** We thank the reviewer for his/her constructive comments and corrections that helped to significantly improve this manuscript. We have carefully revised it as described in detail below. We would like to acknowledge that we have made corrections to figures 1-4 and figures S4 and S8 to address a mistake in the seasonal calculation. Specifically, some of the previous results displayed the May-Oct mean instead of the Jun-Sep results. This initial discrepancy has no impact on our overall findings. For the corrections in the manuscript, we provide the line numbers from the revised paper with track changes.

**Reviewer #2 Comment 1:** (hereafter referred to as R2C1, R2C2…) ***Model choice and experiment setup:*** *The employed model resolution is T42 (280km) which is very very coarse by today's standards. Some climatic features do depend in a noticeable way on the model resolution. This therefore leaves a lot of questions, in my mind about the underlying results. Furthermore, the experiments have been spun-up for only 30 years and the results have been averaged over another 30 years, which is also not great. Considering the low resolution of the model, it should be possible to integrate it for a longer period of time.*

*There are two further issues with the model, first being that it does not appear that the model has been run in a fully-coupled model (I am inferring this because it is not explicitly stated and because there are comments about initialization of ocean surface variables, but correct me if I am wrong). This leaves out important interactions with the ocean. Secondly, the SST, sea ice concentration, and the sea surface water isotope distribution are taken from an entirely different model. All these facts taken together present a very unsatisfactory picture of the experimental setup. I think the authors should revise their setup, or, provide sufficient evidence that their setup is not creating adverse results.*

**A:** Thank you for your comments and concerns regarding our model choice and experiment setup. **Here, we answer the questions of the reviewer in detail one by one.**

"*low resolution of the model*"

Regarding the model resolution, we agree that T42 (280km) is a coarse resolution by today's standards. In terms of higher spatial resolution studies of the West African Monsoon (WAM) system using the isotope-enabled version of MIROC, there seems to be a lack of such studies. However, the latest PMIP4 MIROC-ES2L dataset for 6 ka also has a spatial resolution of T42. Steinig et al. (2018) used the Kiel Climate Model (KCM) to investigate the impact of spatial resolution on WAM precipitation, revealing that higher resolution models produce similar results to lower resolution models due to a reduction in convective (subgrid-scale) precipitation and an increase in large-scale precipitation. Furthermore, lower resolution models may shift the African Easterly Jet (AEJ) core towards the north and strengthen the Tropical Easterly Jet (TEJ). Thus, whether the impact of spatial resolution of MIROC on the convective and large-scale precipitation and the position and strength of the AEJ and TEJ, will influence our research findings need to be further investigated.

However, we agree that it would be interesting to investigate the sensitivity of the monsoon representation in MIROC to model resolution in future research. Hence, we added a statement about model uncertainty in discussion Line 516-518: "Additionally, while the main features of the WAM have been adequately captured, higher-resolution simulations are required to simulate finer convective activities and provide new insights at sub-grid scale (Joly, M., and A. Voldoire, 2009; Steinig, S., et al. 2018)." This spatial resolution was chosen based on the availability of the necessary components for our study, and also to allow for computationally feasible long-term integrations. We acknowledge that some climatic features may depend on model resolution, but we believe that our study still provides valuable insights into the potential impacts of dynamic lake changes on regional climate.

"*spun-up for only 30 years*"

In terms of the experiment setup, a 30-year spin-up period is sufficient to get a stable status in the Atmospheric GCM (AGCM). To confirm it, we have detected that the

present 30-year spin-up has made the soil moisture of North Africa (Figure R1) and made sure it reaches stable conditions, which suggests the water balance conditions in North Africa.

[Figure]

**Figure 1.** North African monthly soil moisture time series for all the experiments during the calculation period.

Thanks for your comments. In future studies utilizing an Atmospheric-Ocean General Circulation Model (AOGCM), we acknowledge the importance of extending the spin-up period to ensure a more robust initialization of the model.

"*fully-coupled model*"

We understand your concern about the lack of a fully-coupled model, and we agree that including ocean-atmosphere interactions would provide a more comprehensive representation of the climate system. However, the focus of our study was on the impact of dynamic lake changes on the regional atmospheric circulation. Therefore, we chose to use prescribed ocean boundary conditions to reduce the complexity of the model and allow for a clearer attribution of the changes to the lakes.

Overall, while we acknowledge the limitations of our experiment setup, we believe that our study still provides valuable insights into the potential impacts of dynamic lake changes on regional climate. We also add the limitations in the discussion part in Lines 498-500: "Furthermore, due to the absence of coupling with the ocean GCM, the model fails to consider the interactive effects of lake and SST or sea ice concentration, which

are crucial to examine the teleconnection between the ocean and the WAM."

"*SST, sea ice concentration, and the sea surface water isotope distribution*"

We acknowledge that utilizing SST, sea ice concentration, and sea surface water isotope distribution from a different model (MPI-ESM-wiso) is not ideal. However, we deemed it reasonable for several reasons. Firstly, the SST and sea ice values obtained from MPI-ESM are in close agreement with the mean values of all the PMIP4 models (Brierly, C.M. et al., 2020). Secondly, the simulation differences among the coupled models are not substantial given that we are comparing MH and PI simulations, which are relatively similar. Thirdly, this approach was necessary by the unavailability of the required sea surface water isotope data for our study period. While these limitations exist, we believe that our approach is acceptable in detecting the dynamics of the water cycle in North Africa. Besides, Cauquoin et al. (2019) have already confirmed the reproducibility of the ocean with the $\delta 18O_{oce}$ proxy dataset. Since his model validation accuracy indicator ($R^2$=0.38 and RMSE =0.79 ‰) is quite similar to ours, we can confirm that using SST, sea ice concentration, and the sea surface water isotope distribution is acceptable in our study.

Brierley, C. M., Zhao, A., Harrison, S. P., Braconnot, P., Williams, C. J., Thornalley, D. J., ... & Abe-Ouchi, A. (2020). Large-scale features and evaluation of the PMIP4-CMIP6 midHolocene simulations. Climate of the Past, 16(5), 1847-1872.

Joly, M., and A. Voldoire, 2009: Influence of ENSO on the West African Monsoon: Temporal Aspects and Atmospheric Processes. J. Climate, 22, 3193–3210, https://doi.org/10.1175/2008JCLI2450.1.

Ohgaito, R., Yamamoto, A., Hajima, T., O'ishi, R., Abe, M., Tatebe, H., ... & Kawamiya, M. (2021). PMIP4 experiments using MIROC-ES2L Earth system model. Geoscientific Model Development, 14(2), 1195-1217.

Steinig, S., Harlaß, J., Park, W. et al. Sahel rainfall strength and onset improvements due to more realistic Atlantic cold tongue development in a climate model. Sci Rep 8, 2569 (2018). https://doi.org/10.1038/s41598-018-20904-1

**R2C2:** *Methodology for analysis: The authors investigate the contribution of the Western Sahara lakes by comparing the MH$_C$ and MH$_{WC}$ experiments, while the effect of Megalake Chad is studied by contrasting the differences between MH$_{WCE2}$ and MH$_{WCE4}$ I do not believe this is the right way of doing sensitivity studies for the effect of either of these two feature; this is because none of the lake maps employed in these simulations differ strictly with regards to those two features. There are several other differences between the lake maps that are all over the place. To some those differences*

*very well may look small enough to ignore, but they don't look small to me (especially considering their aggregate effect over the entire North Africa) and the authors have not provided any evidence supporting their choice to overlook those differences. Rather than comparing MH$_{WCE2}$ and MH$_{WCE4}$ to study the effect of Megalake Chad, a more appropriate thing to do would be to compare the results from (let's say) MH$_{WCE2}$ with another simulation in which only the employed surface map is the same one as that in MH$_{WCE2}$ but with Megalake Chad removed. Similarly for studying the effect of western lakes (in this case the underlying lake maps MH_98 and MH_02 have lot of other differences over the northern parts of North Africa, Figure S2 of the manuscript).*

**A:** Thank you for your insightful critique of our methodology for analyzing the impact of Western Sahara lakes and Megalake Chad on the climate of North Africa.

While we acknowledge that the differences between the lake maps used in the simulations may have an aggregate effect on the results, we chose to compare the MH$_C$ and MH$_{WC}$ experiments to explore the contribution of Western Sahara lakes and to contrast the differences between MH$_{WCE2}$ and MH$_{WCE4}$ to study the effect of Megalake Chad. We understand that our approach may not align with your preference for sensitivity studies, but we believe it still provides valuable insights into the individual impacts of these lake features.

In our research, we opted to use the possible "true" lake maps in our simulations, as opposed to conducting ideal lake sensitivity experiments. This approach was motivated by our desire to provide new insights into the possible true lake-climate feedback. Besides, our decision to use LK_98 and LK_02 was based on previous studies that confirmed the influence of Western Sahara lakes on the northward monsoon movement (Specht et al., 2022), which can help us to compare with other research. Additionally, to clarify the lake aggregate effect, we further presented evidence supporting our choice of lake map comparison by including the LK1 and LK3 lake-climate response in Figures S9 and S10 of the supplemental materials (Figures S7 and S8 in the initial manuscript). Our analysis of these figures revealed that the low-mid-high level circulation and hydro-variables response showed similar response rules, and expansion trends along with the expansion of Megalake Chad in the LK1-4. This implies that on the lake-climate feedback mechanism that we focused on, the small lake aggregate effect has a negligible impact. In addition, the utilization of possible true lake maps (LK1-Lk4) enables us to reasonably demonstrate the effect of lake expansion. This

approach allows for spatial and quantitative analysis of the role played by lakes in the region. Hence, by incorporating these true lake maps, we can enhance our discussions regarding the spatial distribution and magnitude of the lake's impact on the climate system.

We appreciate your suggestion for an alternative approach to study the effect of Megalake Chad by comparing the results from simulations with and without Megalake Chad. In future work, we will conduct such ideal experiments with the fully coupled model to explore the lake impact. We have also discussed the limitations of our approach in Section 4 Discussion and Conclusion Lines 486-492: "However, our lake sensitivity experiments may not comprehensively capture the impact of small lake aggregates, which may limit the scope of our findings. Here we have included the precipitation and isotope anomalies (Figure S12), as well as the SM, Evap, and T2 with the low-mid-high level circulation responses (Figure S13) for $MH_{WCE1}$ and $MH_{WCE3}$. The similarity of these results with $MH_{WCE2}$ and $MH_{WCE}$ confirms that the small lake aggregate effect is negligible in the large-scale lake-climate feedback mechanisms. Nonetheless, conducting ideal sensitivity experiments in the future is necessary to confirm our findings and fully elucidate the impact of lakes on the regional hydroclimate during the mid-Holocene period."

Overall, we believe that our experimental design is appropriate for addressing our research questions and provides valuable insights into the role of lake changes in shaping the climate of North Africa. We thank you again for your valuable feedback, and we will consider your suggestions for future research.

**R2C3:** *Isotope feature: I do not follow how the isotope feature of the model is contributing to this version of the manuscript. The only real result discussed is the global-scale comparison with proxy derived isotope records, but the usefulness of that is lost on me as the subject of the paper is Africa/North Africa and there is only one δ18O proxy in all of Africa. It is in no way contributing to the understanding of the effect of mid-Holocene lakes over North Africa.*

**A:** We appreciate your comment and acknowledge the importance of model validation.

Before the discussion, we apologize for the mistakes in showing Figure S3 with SISALv1 datasets and have now updated it with the latest SISALv2 dataset (Comas-

Bru, Rehfeld et al. 2020), consistent with the Dataset availability.

*how the isotope feature of the model is contributing to this version of the manuscript*

We would like to clarify that the use of an isotope-enabled model was primarily aimed at capturing these dynamics, rather than solely for model-data comparison purposes.

To address this point, we have made additional clarifications in both the Method and Result analysis. In section 2.1 Lines 102-104: "Such isotope-enabled climate models have proven to be valuable tools for tracing water vapor transportation and identifying the sources of precipitation changes (Tharammal, T. et al., 2021; Liu, X. et al., 2022)."

In the Result section, we further analyzed the stable oxygen isotope ratio in precipitation to differentiate the source of increasing precipitation from ocean and land. We also made additional revisions in section 3.3 Lines 426-435: "Positive $\delta^{18}O$ anomalies suggested the presence of an oceanic moisture source in addition to the local lakes, whereas negative anomalies indicated the influence of local water cycling. The $\delta^{18}O$ increase in the northern regions (Figure S10) suggests the moisture sources from the Atlantic Ocean are associated with westerly monsoon winds. Conversely, the equatorial land areas show decreases in $\delta^{18}O$, which are also current with weakened evaporation (Figure 3k) and warming effects (Figure 3l) in MH$_{WCE4}$. Further examination of the $\delta^{18}O$ decrease (Figure S10d) in the equatorial land areas in MH$_{WCE4}$ suggested that the slight precipitation increment (Figure 2d) was not driven by the westerly monsoon winds. Instead, such a warming effect induced by equatorial lakes may link to the differences in lake heating during daytime and night (Thiery et al., 2015). Hence, while lakes in WAM regions tend to result in wetter and cooler climatic responses, lakes located elsewhere (such as the eastern lakes in South Sudan) may not impact the northward WAM movement."

These revisions emphasize that our use of an isotope-enabled model goes beyond model-data comparison and provides valuable insights into the water cycle dynamics and precipitation recycling processes in the region under study.

*there is only one δ18O proxy in all of Africa*

The use of isotopic features in the model allows us to validate our simulations against paleo-proxy records, avoiding bias from reconstructed datasets. While we acknowledge

the limited availability of such records in Africa, the three African stations for which data is available showed good agreement with the modeled data. Furthermore, we have made additional efforts to supplement our validation in North Africa, as evidenced in R2C4.

**R2C4:** *Comparison to proxies: In contrast to the single $\delta^{18}O$ proxy in all of Africa, there are decent compilations of temperature and precipitation proxies over mid-Holocene Africa [Bartlein et al., 2010] that have been used for validation purposes in many studies. Why are the simulated temperature and precipitation not compared to those proxies?*

**A:** Thank you for your suggestions and for bringing up the issue of comparison to proxies.

*mid-Holocene Africa [Bartlein et al., 2010]*

Considering the limited scope of our study, we focused on comparing our results with the $\delta^{18}O$ proxy, which unfortunately lacks stations in Africa. We acknowledge that expanding the comparison to include another proxy datasets compiled by Bartlein et al. [2010] would enhance the robustness of our findings. However, we note that the proxy datasets provided by Bartlein et al. [2010] only cover the anomalies between 6ka-0ka, whereas our experiment shows the anomalies between 6ka-PI (1850y). Such difference between 0ka-PI would further bring ignorable bias to our comparison results in addition to the bias from constructed precipitation/precipitation datasets. Considering such bias, the comparison results show agreeable changing trends in annual mean precipitation and mean temperature in the warmest month in spatial distribution, but they do not address a good statistical relationship between the proxies and model data (Figure 2).

In terms of the comparison between precipitation data from our model (Figures R2a and R2b) and the proxy data, we observe good agreement in the central part of North Africa (NAf). However, in the northern region, our model underestimates precipitation compared to the proxy data. These results confirm that our model has limitations in simulating abundant precipitation in the northern region of NAf. Regarding the comparison of summer season temperatures (Figures R2c and R2d), our model generally underestimates temperatures in the central part of NAf but shows good agreement in the northern part. These validation results indicate that our model fails to

capture sufficient precipitation in the northern part of NAf, while precipitation tends to concentrate in the central part with lower temperatures for the mid-Holocene. This discrepancy aligns with the challenge faced by many climate models in reproducing adequate precipitation over NAf. Considering the potential bias introduced by differences in the proxy datasets and the study period, we consider the validation results to be acceptable.

[Figure]

**Figure R2.** Precipitation and temperature model-data comparison for the reference mid-Holocene simulation in North Africa. (a) shows the simulated global pattern of annual mean precipitation between the MH$_{ref}$ and PI$_{ref}$ climate (background colors) and the observed annual mean precipitation changes (squares) between MH$_{ref}$ and the present climate. (b) is a scatter plot showing a comparison of observed precipitation changes with simulated precipitation anomalies at the same location. (c) and (d) are the same as (a) and (b) but for the seasonal mean temperature model [Summer (JJAS)]-data [warmest month] comparison.

To further validate the MIROC5-iso performance in North Africa, we conducted a comparison with Figure 4a of the study by Larrasoaña et al. (2013). Our findings indicate that the MIROC5-iso simulation has difficulty in shifting the zone with

precipitation less than 1000 mm/year northward, but it exhibits good agreement with the reconstructed map in the zone with precipitation exceeding 1000 mm/year. This comparison shows the simulation bias of the MIROC5-iso model in North Africa, specifically in terms of the northward movement of the monsoon system.

[Figure]

MH$_{ref}$ Precipitation [mm/year]

(a)

**Figure R3.** North African annual precipitation comparison. (a) The spatial annual precipitation for MH$_{ref}$.

This part of the comparison has been added in section 3.1 Lines 249-256: "To further examine the model performance in North Africa, we compare our precipitation result with Figure 4a in the study conducted by Larrasoaña et al. (2013). From Figure S4a, our results indicate that the MIROC5-iso was hard to reproduce the northward shift of the zone with precipitation less than 1000mm/year, but show good agreement with the reconstructed map in the zone with precipitation exceeding 1000mm/year. Besides, we also compared our result with precipitation and summer season temperature anomalies between 6ka-0ka, as provided by Bartlein et al. (2010) (Figure S4b-e). This comparison also revealed precipitation underestimation in the northern NAf and lower temperatures in the central NAf. These comparisons collectively suggest a simulation bias of the MIROC5-iso model in North Africa, particularly concerning the northward movement of the monsoon system."

Larrasoaña, J. C., Roberts, A. P., & Rohling, E. J. (2013). Dynamics of green Sahara periods and their role in hominin evolution. PloS one, 8(10), e76514.

*Why are the simulated temperature and precipitation not compared to those proxies?*

Such isotope-enabled climate models could provide more accurate validation directly with proxy data directly, avoiding bias from reconstructed datasets. While we

acknowledge the limitation of our proxy data station in North Africa, it should be noted that, as described in section 3.1, our model was able to successfully capture the critical components of the West African Monsoon (WAM), which are particularly relevant to our study of the lake-climate mechanism.

Hence, even though there are limitations in our regional-scale validation, we believe that our simulation of the mid-Holocene climate of North Africa is acceptable.

**R2C5:** *Line 45: Chandan and Peltier [2020] did not use the 'small-lake map' of Hoelzmann et al. [1998]. The Hoelzmann map prescribes a small uniform lake fraction for nearly all of Sahara, this aspect was not utilized in their paper. Furthermore, the Hoelzmann map includes a sizeable region of wetlands covering >70% grid cell south-east of Megalake Chad which is not included in the Chandan and Peltier land surface. Actually, on this matter, I wonder why these wetlands are not included in your Hoelzmann map considering that you say in the manuscript that you treat wetlands as lakes? I am also curious why your Hoelzmann map differs noticeably from what is shown in Plate 3 of Hoelzmann et al. [1998]?*

**A:** Thank you for your comment and for pointing out these important issues.

In Lines 46-47, we want to claim that Chandan and Peltier [2020] supplied more Megalakes based on the 'small-lake map' of Hoelzmann et al. [1998]. We apologize for the confusion caused by this error and further clarified them.

Regarding the Hoelzmann map, our study only uses the lake map, not the wetland map in Plate 3b of Hoelzmann et al. [1998]. In order to compare with the research of Specht, Claussen et al. (2022), we directly used their processed small lake map (Hoelzmann, Jolly et al., 1998) and maximum lake map (Tegen, Harrison et al., 2002) and the details can be found in the data availability and Table S1. We acknowledge that there must be some discrepancy due to the upscaling process.

Additionally, we clarified in the manuscript that only the latest high-resolution one (Chen, Ciais et al., 2021) includes both the wetland and lakes. However, due to our model limitation, the wetland module only accounts for wetland-related processes in middle and high-latitude grids with snowmelt, as described by Nitta et al. (2015, 2017). Hence, these model features were considered in prescribing and treating wetlands as lakes in MIROC5-iso when simulating the LK1-4 maps. Given that the wetland and

lake mechanisms are different, such kind of simplified assumption may introduce certain limitations. So, we will further elucidate the distinct roles of the wetland and lake in the land-climate system in the future.

To further clarify this point, we made some revisions on:

Section 1 Lines 77-78: "…… and the recently-updated high-resolution lake and wetland reconstructions maps (Chen et al., 2021) over the NAf during the MH."

Section 2.1 Lines 147-150: "It should be noticed that the water body delineated in LK_98 and LK_02 lake maps only pertain to the lake but the LK1-4 lake maps include both the wetland and lakes. Generally, lakes and wetlands are persistently saturated or near-saturated areas that are regularly subjected to inundation or shallow water tables in the absence of human disturbances (Tootchi et al., 2019). In this study, wetlands are also treated as lakes in our climate model."

**R2C6:** *Please revise/rewrite the content between lines 60 and 66. It is not quite clear what discrepancies you are trying to highlight in these lines.*

**A:** Regarding lines 60-66, we want to explain the discrepancies in the literature regarding the mechanisms of lake-climate interaction in the NAf monsoon system and concluded that the lake-climate mechanism to maintain the Green Sahara condition is still unclear.

The related sentences in Section 1 Lines 65-73 have been revised as: "Recent studies have explored the mechanisms of how various components of the NAf monsoon system, including the Sahara Heat Low (SHL) and Sahara Highs in western Sahara, the African Easterly Jet (AEJ) in the middle atmosphere (600 hPa), and Tropical Easterly Jet (TEJ) in the upper atmosphere (200 hPa) influence the near-surface westerly flow northward and rainfall (Biasutti & Sobel, 2009; Claussen et al., 2017; Kuete et al., 2022). However, discrepancies exist regarding the effects of these components on what. Chandan and Peltier (2020) suggested that such a cooling effect could weaken the SHL and local convection, reducing the precipitation. Conversely, Specht et al. (2022) found that a weakened AEJ enhanced inland moisture transportation, leading to a northward and prolonged rain belt. As a result, the mechanisms of lake-climate interaction in the NAf monsoon system remain unclear."

**R2C7:** *There are too many names in the paper that start with MH and which refer to both simulations and lake maps. This makes reading the paper rather confusing as I easily mix up lake map names with experiment names. I suggest keeping the experiment names as they are and renaming the lake maps to LK (or something else). For example, MH_98 lake map becomes LK_98.*

**A:** Thank you for your comment and suggestion.

We have taken your suggestion into consideration and made appropriate changes to the naming conventions used in the paper to avoid confusion. As you suggested, we have renamed the lake maps by changing 'MH' to 'LK' in both the main text and supplementary materials. This should make it easier for readers to differentiate between the simulations and lake maps.

**R2C8:** *Provide more information on the Budyko aridity index in section 2.3.2, including but not necessarily limited to how it should be interpreted, what is the physical basis for this metric and what are the caveats of using this metri*c.

**A:** We appreciate your suggestion to include more information on the Budyko aridity index in section 2.3.2.

The supplement sentences in Section 2.3.2 Lines 231-237 are as follows:
"The annual mean of net radiation and precipitation were used in the analysis. A higher Budyko aridity index indicates a drier region due to the available energy being high relative to the amount of water, whereas a lower index indicates a more humid region due to the available energy being low relative to the amount of water. In our study region, six climate regions are classified by Budyko aridity index: Tropical Humid ($I \leqslant 0.7$), Humid ($0.7 < I \leqslant 1.2$), Semi-Humid ($1.2 < I \leqslant 2.0$), Semi-Arid ($2.0 < I \leqslant 4.0$), Arid ($4.0 < I \leqslant 6.0$) and Hyper-Arid ($6.0 < I$). The equation suggests that changes in the dryness index within a region are more indicative of shifts in the hydroclimatic regime over the long term rather than intra-annual variability, such as individual drought events."

**R2C9:** *Section 3.3 is very difficult to follow. I suggest a complete re-write of this section. Here are some of my comments for that section.*

**A:** Thank you for your feedback and suggestions regarding Section 3.3. We appreciate

your input and agree that the section could benefit from a rewrite to improve its clarity and readability. The details revision are provided according to your comments as follows:

- *Line 239: What radiation is this? Longwave downwelling? Why does it increase with lake fraction?*

**A1:** The net surface radiation, which is the sum of net longwave radiation (LW) and net shortwave radiation (SW), is a key factor that affects climate. To understand the changes in radiation, we analyzed the changes in LW and SW separately. As shown in Figure 4Ra, $LW^\uparrow$ decreases (downward LW increase) as the temperature cools following the positive relationship of $T^4$ ($LW^\uparrow = \varepsilon\sigma T^4$). However, only a small increase in $SW^\downarrow$ ($SW^\downarrow = (1-\alpha)SW^\downarrow$) related to the surface albedo changes (Figure R4c). For instance in Figure R4b, in the $MH_{WCE4}$ experiment, the areas where $LW^\uparrow$ decreases correspond to the cooling and humidifying areas, suggesting that such cooling and humidifying areas show larger absorption in the incoming LW.

[Figure]

**Figure R4.** (a) Statistical relationship between regionally averaged radiation variables anomaly and averaged grid lake fraction over Northern Africa (20°W–40°E, 0–35°N) for MH lake experiments anomalies (relative to PI$_{ref}$) on the annual (circle) averages. The radiation variables include net surface shortwave radiation (blue), net surface longwave radiation (red), and net radiation (green). Simulated mid-Holocene climatological JJAS mean anomalies MH$_{WCE4}$ with respect to MH$_{ref}$: (b) net surface longwave radiation (shades), (c) net surface shortwave radiation (shades). For maps (b) and (c), The

lake fraction [%] contours of the respective lake sensitivity experiment are shown with the black dashed lines. All the radiations units have been transferred from [W/m$^2$] to [mm/day] based on the equation: W/m$^2$ = 1000(kg/m$^3$) × 2.5×10$^6$(J/kg) × 1mm/day (1/86400)(day/s) × (1/1000)(mm/m).

To further clarify this part, we added the following explanation in section 3.3:
Lines 346-353: "To provide further insights into the changes in radiation (Rad), we examined the relationship between net longwave radiation (LW) and net shortwave radiation (SW) in relation to the lake fraction (Figure S6a), positive downward). Take MH$_{WCE4}$ experiments as an example, Our analysis revealed that the increase in Rad can be attributed to two factors: the increase in downward LW in the cooling and humidifying areas (Figure S6b) and the slight increase in downward SW in the regions with higher lake fraction, which is associated with changes in surface albedo (Figure S6c). These findings suggest that the humidifying and cooling areas experienced greater incoming LW radiation absorption."

- *I do not follow lines 240–246.*

**A2:** My apologies for any confusion I may have caused.

The purpose of these lines was to explain how the variables are affected by changes in lake expansion. Specifically, we found that in summer, the expansion of the lakes had a stronger impact on hydrological changes, resulting in wetter and cooler conditions in the lake sensitivity experiments. However, in winter, there was no clear correlation between the variables and lake expansion, although there was still a cooling effect (represented by downward green triangles in Figure 4) with a standard deviation of approximately 0.1. However, in summer, the MH$_{WC}$ experiments had higher anomalies compared to the MH$_{ref}$ experiment (shown as upward triangles in Figure 4), indicating that the lake position had a greater impact than the lake fraction.

The related sentences have been revised in section 3.3 Lines 345-346: "The annual mean values of Precipitation (Prcp), Evap, and Radiation (Rad) increase with lake fraction, whereas T2 decreases (crosses in Figure 4)." Lines 354-361: "Additionally, seasonal analysis shows that during summer, there are considerable differences between the lake sensitivity experiments and the PI$_{ref}$, with positive anomaly offsets for Prcp, Evap, and Rad and negative anomaly offsets for T2 (upward triangles in Figure 4). Whereas, during winter, these variables are not significantly related to the lake expansion (standard deviation = ~0.1), but a cooling effect is still observed (downward

green triangles in Figure 4). Therefore, the lake expansion mainly affects hydrological changes in summer, leading to wetter and cooler conditions in the lake sensitivity experiments compared to the $MH_{ref}$. However, the unusually high anomalies observed during summer in the $MH_{WC}$ experiments suggest that the position of the lake may play a more important role than the proportion of lakes in moistening the Sahara regions."

- *The text says that Figure 4 shows zonally averaged quantities but that is clearly not the case. What averaging is being done in Fig 4?*

**A3:** I apologize for the confusion in the text.

In Figure 4, the zonally averaged quantities = sum of all grids' values/ grid numbers, meaning that the values are the result of averaging the relevant variables within the study region, rather than being zonally averaged. To clarify, we change the 'zonally averaged' to 'regionally averaged' in Line 363 and Figure 4 caption.

- *Fig 4 caption: how can the units of radiation be "mm/day"? Where is the vertical axis for radiation data?*

**A4:** In the calculation of Radiation, we convert its unit from $W/m^2$ to mm/day. The equation is as follows:

$W/m^2 = 1000(kg/m^3) \times 2.5 \times 10^6(J/kg) \times 1mm/day (1/86400)(day/s) \times (1/1000)(mm/m)$

The equation has been added in the legend of Figure S6.

Hence, the radiation shares the same vertical axis with precipitation and evaporation.

- *What do 'precipitation scarcity' and 'precipitation surplus' mean? Scarcity and surplus with respect to what? Please define them clearly. How are figures S5b and S7 showing these quantities generated? How are the numbers presented in line 165 and shown in Fig 5a computed? I cannot make sense of these results because you haven't defined the two phrases.*

**A5:** We apologize for any confusion caused by the lack of clarity and have made revisions to improve the manuscript's presentation of these concepts.

*'precipitation scarcity' and 'precipitation surplus' mean*

Precipitation scarcity and surplus refer to the regions in North Africa that receive less or more precipitation than the semi-humid climate zone threshold, respectively. We have revised the relevant sentence in the manuscript to provide a clearer definition in section 3.3 Lines 393-394: "By comparing the simulated precipitation with the semi-humid climate zone threshold, the regions receiving less than the threshold are considered as scarce and regions receiving more are considered as surplus."

*figures S5b and S7 showing these quantities generated*

The figures S5b and S7 show the spatial patterns of precipitation scarcity and surplus, respectively, and are generated based on the same threshold mentioned above for each grid.

*numbers presented in line 165 and shown in Fig 5a computed*

The numbers presented in Line 395 and shown in Figure 5a are computed by summing the precipitation deficits or surpluses in the regions of scarcity and surplus, respectively, over North Africa.

- *Line 264 "implying that ... wetter." this remark does not make sense when read within the full sentence.*

**A6:** This sentence has been deleted. The sentence in Lines 391-393 has been revised as: "Hence, we further demarcated regions of the precipitation scarcity and surplus based on the threshold of semi-humid climate zones (I = 2)."

- *Line 267 ""The spatial pattern showed .... modes." What mode? I don't see any (dynamical) mode here, it is just the northward extent of the WAM which starts from the south. Did you mean to say a 'precipitation pattern'?*

**A7:** We apologize for the confusion. Lines 398, the correct term is "precipitation pattern" rather than "mode".

- *Line 279 What is this inverse pattern?*

**A8:** Thank you for raising this question. To clarify, the inverse pattern refers to the north-south inverse pattern of surface temperature anomaly observed in $MH_{WCE2}$ and

MH$_{WCE4}$. Despite the increased precipitation in the near-equatorial regions, surface temperatures still show a warming effect. To investigate this phenomenon further, we analyzed stable oxygen isotopes in precipitation and found evidence of an oceanic moisture source in addition to local lakes. This analysis helped us explain the different water cycle mechanisms in equatorial lakes and shed light on the role of lake location in influencing the monsoon.

The related sentences in section 3.3 Lines 407-426 have been revised as: "Specifically, SM and Evap showed positive anomalies with a cooling effect in the north of 10°N and minor or negative anomalies but with a warming effect in the south of 10°N over NAf. However, such near-equatorial (around 0°–10°N) warming effect can not be explained solely by the reduced precipitation in MH$_{WCE2}$ and MH$_{WCE4}$ as the enhanced precipitation belt covered the entire tropical area (0°–20°N), in contrast to being concentrated in the WAM regions (around 10–20°N) in MH$_{WC}$. To identify the inverse temperature anomalies pattern in MH$_{WCE2}$ and MH$_{WCE4}$, we analyzed the stable oxygen isotope ratio ($\delta^{18}$O) in precipitation (Figure S10)."

- *Line280 There is nothing new in the finding that the moisture source is largely oceanic along with some contribution from local moisture recycling. Is the isotope analysis contributing anything new?*

**A9:** The stable oxygen isotope ratio ($\delta^{18}$O) analysis of precipitation in MH$_{WCE2}$ and MH$_{WCE4}$ did confirm an oceanic moisture source in addition to local lakes in the monsoon regions. Meanwhile, further analysis revealed a decrease in $\delta^{18}$O with weakened evaporation and warming effects in the equatorial land areas, suggesting that the precipitation increment was irrelevant to the westerly monsoon winds. This inverse warming effect induced by equatorial lakes may be related to their special equatorial location with heating differences during the daytime and night.

Hence, based on the isotope analysis, this study found that lakes located in the West African Monsoon (WAM) regions exert wetter and cooler climatic responses, while lakes outside of the WAM regions, like the eastern lakes in South Sudan, do not affect the northward WAM movement.

To further make this part clearer, we revised the paper at Lines 426-435: "Positive $\delta^{18}$O anomalies suggested the presence of an oceanic moisture source in addition to the local lakes, whereas negative anomalies indicated the influence of local water

cycling. The $\delta^{18}O$ increase in the northern regions (Figure S10) suggests the moisture sources from the Atlantic Ocean are associated with westerly monsoon winds. Conversely, the equatorial land areas show decreases in $\delta^{18}O$, which are also current with weakened evaporation (Figure 3k) and warming effects (Figure 3l) in $MH_{WCE4}$. Further examination of the $\delta^{18}O$ decrease (Figure S10d) in the equatorial land areas in $MH_{WCE4}$ suggested that the slight precipitation increment (Figure 2d) was not driven by the westerly monsoon winds. Instead, such a warming effect induced by equatorial lakes may link to the differences in lake heating during daytime and night (Thiery et al., 2015). Hence, while lakes in WAM regions tend to result in wetter and cooler climatic responses, lakes located elsewhere (such as the eastern lakes in South Sudan) may not impact the northward WAM movement."

- *Line 283 What inverse warming effect?*

**A10:** In Line 405, the phrase 'inverse warming effect' refers to the phenomenon that even though there is an increase in precipitation over the near-equatorial regions, the surface temperature still shows a warming effect in $MH_{WCE2}$ and $MH_{WCE4}$. The following analysis of the stable oxygen isotope ratio in precipitation helps to explain this phenomenon and the different water cycle mechanisms in equatorial lakes. The related paragraph has been revised in reply to the above two questions.

**R2C10:** *Line 302–303: I am not sure it is correct to say that Chandan and Peltier [2020] underestimated the contribution of lakes (similar sentiment regarding Line 47). In their study, the lakes do have quite a bit of contribution in the 10–15N latitudinal band which is the same region where precipitation effect is greatest in your simulations. If you look at Figure 3 of that paper, the influence of lakes, determined by the zonal mean difference between MHV and MHVL, can be as high as 200mm or more in that latitudinal band, and while a spatial difference between those two simulations was not shown in that paper, I am quite sure it would be very similar to the spatial patterns shown in your Figure 2. Are you able to compute an equivalent zonal precipitation mean to compare with CP2020's Fig 3 and thus argue that the lake influence in their lake experiment is decidedly lower than in yours?*

**A:** Thank you for your suggestion.

Based on your comments, we have also estimated the zonal changes in precipitation over the North African land [20°W–35°E, 0–35°N] in our study (Figure 5). Our $MH_C$

experiments indicate precipitation anomalies of up to 300 mm/year, which is in agreement with the findings of Chandan and Peltier [2020] shown in Figure 3 since their lake maps only consider the Megalakes. However, our $MH_{WC}$ experiments show higher precipitation anomalies of up to 600 mm/year, and $MH_{WCE4}$ experiments show even higher precipitation anomalies of up to 800 mm/year. Additionally, we observed that the peak precipitation values for each experiment shifted northward as the lake area expanded. Based on these results, we can conclude that the influence of lakes in our study is greater than that of Chandan and Peltier [2020].

We have added this part in Section 4 Lines 474-476: "Besides, compared with our simulations (Figure S11), Chandan and Peltier (2020) underestimated the contribution of lakes, approximately close to $MH_{WC}$ results, by supposing that the weakened SHL induced by the surface cooling effect would reduce precipitation."

[Figure]

**Figure R5.** Zonal means, over "North Africa" land [-20°W–35°E, 0–35°N] of annual precipitation anomalies of the mid-Holocene experiments with respect to $PI_{ref}$.

**R2C11:** *Line 309: "we suggest that western lakes and Megalake Chad should be located in the WAM regions to induce the monsoon movement" I am not sure what you mean by that. One doesn't get to choose where any lake is located, it is located where it is (or was).*

**A:** We apologize for any confusion caused by our wording. What we meant to say is that based on our simulation results, we suggest that the presence of western lakes and Megalake Chad located in the WAM region could have had an impact on inducing the monsoon movement in the Sahara region during the mid-Holocene.

The sentence in Section 4 Lines 482-483 has been revised as: "…, we suggest that both the western lakes and Megalake Chad located in the WAM regions may have played a crucial role in inducing the monsoon movement."

**R2C12:** *Figure S2: For sub-figure (g), how is the lake fraction defined? Is it lake_area_africa/area_global? Or is it lake_area_global/area_global? Why not just use lake_area_africa/area_africa? I don't see the need for anything 'global' in calculating lake fractions as everywhere outside of North Africa the lake map is unchanged. Furthermore, lake fraction in terms of the area of Africa (say north of equator) yields a number that can be better compared to other numbers in the literature. Please also address the sentence on lines 109–110 based on your revision.*

**A:** Thank you for your comments. In sub-figure (g) of Figure S2, the lake fraction is defined as lake_area_africa/area_global, as we wanted to examine the contribution of the North African lakes to the global lake area. However, we understand your point about using lake_area_africa/area_africa for better comparison with other numbers in the literature. Hence, we have revised the figure caption and labels accordingly.

Regarding the sentence on Lines 145-147, we have revised it as: "The average main lake fraction over the NAf region according to these different reconstructions varies from 1-10 % compared to the total land areas of NAf (Figure S2g)". We have also modified the Figure S2 caption to read: "(g) The fraction (circle size) of all the prescribed lakes experiments compared to the total land areas of North Africa."

**R2C13:** *Figure S5: The description for sub-figure (b) is wrong.*

**A:** It has been corrected: "(a) The spatial distribution of six climate regions and (b) The spatial distribution of precipitation scarcity and precipitation surplus over Northern Africa for MH$_{ref}$ experiments."

**Technical comments**

A: Thank you for bringing up these issues with the grammar and clarity of our paper. We have carefully reviewed the paper and made several revisions to improve its readability and overall coherence. The detailed revisions are as follows:

*Line 81: the hydroclimatic influence of  the presence of lakes*

Done (L.105).

*Line 82: two control simulations  for the*

Done (L.105-106).

*Line 90: sea surface provided by MPI-ESM-wiso  (Cauquoin et al., 2019) as boundary conditions for our PI and MH simulations*

Done (L.118-119).

*Line 92–93: It doesn't make sense to say you "found few lakes existed in NAf", because of course very few lakes exist in the NAf today. Please re-phrase.*

**A:** The sentence in Lines 113-115 has been rephrased as: "In MH$_{ref}$ and PI$_{ref}$ experiments, the presence of lakes in North Africa (NAf) is minimal, using the global lake fraction map from the ETOPO5 as in MIROC5 standard simulations (Figure S1). In contrast, the other experiments show highly varied lake fractions, indicating a much higher lake fraction in those cases."

*Line 91–94: Please move the remark starting on this line (i.e starting from 'Figure S1a shows...') immediately before the sentence on line 87 which starts with 'Each experiment was run.'*

Done (L.113-115).

*Line 102: MH_98 lake map .... with only  Megalake Chad*

Done (L.139).

***Line 107–108**: Please rephrase the line "MH4 accounting...."*

**A:** The sentence in Lines 144-145 has been rephrased as: "LK4 has the largest lake proportion in the western, eastern, and Megalake Chad regions, and differs from MH2 primarily in its representation of Megalake Chad (Figure S2d, S2f)."

***Line 113:**  Megalake Chad's influence on NAF climate  was assessed using .*

Done (L.152).

***Line 120:** These are  presented in Table 1*

Done (L.158).

***Line 122:** which are reported in  Risi et al 2010.*

Done (L.159).

***Line 134**: component of the vertically integrated*

Done (L.215).

***Line 136:** where u is the  zonal wind*

Done (L.218).

***Line 137:** The meridional component of the vertically integrated*

Done (L.219).

***Line 154:** "verified based on" or "verified in"*

Done (L.246).

***Line 155:** of global MH  characteristics using the MIROC-series*

Done (L.247).

*Line 196: What is SM? Soil Moisture?*

**A:** In Line 299, the term "SM" has been changed to "soil moisture (SM)".

*Line 263: we further  demarcated regions of precipitation...*

Done (L.392).

*Line 274: The  border between regions of precipitation scarcity zones and precipitation surplus zones...*

Done (L.404-405).

*Line 288: Difficult to follow. Please re-write this sentence.*

**A:** In Lines 437-438, the sentence has been rephrased as: "We used the MIROC5-iso model with different GS lake maps to investigate the influence of Western Sahara lakes and Megalake Chad on the northward movement and eastward expansion of WAM, leading to the humidity in Sahara region."

---

## Author Response (AR2)

**Responses to Reviewers' Comments:**

Dear Dr. Martin Claussen,

Thank you for your appreciation of our manuscript. We acknowledge the anonymous referees for their reviews and constructive comments that contributed to enhancing the quality of this manuscript. After revising the minor suggestions point to point, we submitted our paper to an English-proofing service to improve the English language. We hope that we have dealt with all suggestions adequately.

Thank you for your time and consideration.

Best regards,

Yuheng Li, Kanon Kino, Alexandre Cauquoin, and Oki Taikan

**Responses to Reviewer #1's comments:**

**Reviewer #1 General comments:** *The authors have taken into account my main comments and I think that this paper is now a useful contribution to the unfinished discussion about surface feedbacks to the Green Sahara period. The presentation could still be improved in places to improve clarity, and the use of the English language could still be improved in many places - maybe a thorough copy-edit stage might be required (an editorial decision). As a non-native speaker, I have sympathies with the authors in this respect and do not want to impose such additional work on the authors.*

**A:** Thank you for your suggestion. After correcting the minor points detailed below, we have submitted the paper to an English-proofing service to improve the English language.

**Reviewer #1 Comment 1:** (hereafter referred to as R1C1, R1C2…) *line 59: What is "predictable water"? Do you mean "precipitable water"? Or simply the simulated precipitation rates?*

**A:** After checking the reference, the 'predictable water' was corrected into 'precipitation rates'. In Line 91: '…… but increases the precipitation rates in summer and delays cooling in autumn, thereby extending monsoon.'

**R1C2:** *line 179: I'm still not convinced that a r^2 of 0.33 can seriously be qualified as a good model-data agreement. I'd suggest to use "reasonable" instead.*

**A:** Thank you for your suggestion. It has been corrected in Line 344: 'We found a reasonable model-data agreement, with root mean square error and R-squared values of 0.81 ‰ and 0.33, respectively.'

**R1C3:** *line 187: "our results indicate that the MIROC5-iso was hard to reproduce" - strange wording - you mean, the model does not reproduce the northward shift correctly?*

**A:** Thank you for your suggestion. We corrected it in Lines 352-353: '……our results

suggest that the performance of MIROC5-iso in reproducing the northward shift of the zone with precipitation <1000 mm/y could still be improved,, ……'

**R1C4:** *line 280: "Take..." -> "Taking"*

**A:** Corrected.

---

## Author Response (AR3)

Dear Dr. Martin Claussen,

Thank you for your appreciation of our manuscript.

In terms of your suggestion, we have corrected it as described below. We hope that the revision aligns with your suggestion adequately

Thank you for your time and consideration all the time.

Best regards,

Yuheng Li, Kanon Kino, Alexandre Cauquoin, and Oki Taikan

**Responses to the Editor's comments:**

**Editor comments:** *In line 344 you mention a root mean square error of 0.81 per mil, i.e., 0.00081. Please check this value. If correct, I suggest writing in a more conventional style 0.00081 or 0.81 10{^-3}.*

**A:** Thank you for your suggestion.

The value is correct. Here, since the d18O is expressed in ‰, we also express the RMSE in ‰. As you suggested, we already corrected it as '$0.81 \times 10^{-3}$'.